# BNEM: A Boltzmann Sampler Based on Bootstrapped Noised Energy Matching

**RuiKang OuYang***  *ro352@cam.ac.uk*
*University of Cambridge*

**Bo Qiang***  *bqiang@uw.edu*
*University of Washington*

**José Miguel Hernández-Lobato**  *jmh233@cam.ac.uk*
*University of Cambridge*

**Reviewed on OpenReview:** *https://openreview.net/forum?id=ZZktUOU6Pu*

## Abstract

Generating independent samples from a Boltzmann distribution is a highly relevant problem in scientific research, *e.g.* in molecular dynamics, where one has initial access to the underlying energy function but not to samples from the Boltzmann distribution. We address this problem by learning the energies of the convolution of the Boltzmann distribution with Gaussian noise. These energies are then used to generate independent samples through a denoising diffusion approach. The resulting method, NOISED ENERGY MATCHING (NEM), has lower variance and only slightly higher cost than previous related works. We also improve NEM through a novel bootstrapping technique called BOOTSTRAP NEM (BNEM) that further reduces variance while only slightly increasing bias. Experiments on a collection of problems demonstrate that NEM can outperform previous methods while being more robust and that BNEM further improves on NEM. Codes are available at https://github.com/tonyauyeung/BNEM.

## 1 Introduction

A fundamental problem in probabilistic modeling and physical systems simulation is to sample from a target Boltzmann distribution $\mu_{\text{target}}(x) \propto \exp(-\mathcal{E}(x))$ specified by an energy function $\mathcal{E}(x)$. A prominent example is protein folding, which can be formalized as sampling from a Boltzmann distribution (Śledź & Caflisch, 2018) with energies determined by inter-atomic forces (Case et al., 2021). Having access to efficient methods for solving the sampling problem could significantly speed up drug discovery (Zheng et al., 2024) and material design (Komanduri et al., 2000).

Traditional methods for sampling from unnormalized densities include Monte Carlo techniques such as AIS (Neal, 2001), HMC (Betancourt, 2018), and SMC (Doucet et al., 2001), which are computionally expensive. Amortised methods using machine learning techniques have been developed in recent years (Noé et al., 2019; Midgley et al., 2023; Vargas et al., 2023a;c; Berner et al., 2024; Albergo & Vanden-Eijnden, 2024; Rissanen et al., 2025). However, most existing methods for sampling from Boltzmann densities have problems scaling to high dimensions and/or are very time-consuming. As an alternative, Akhound-Sadegh et al. (2024) proposed Iterated Denoising Energy Matching (iDEM), a neural sampler based on denoising diffusion models which is not only computationally tractable but also seems to provide good coverage of modes. Nevertheless, iDEM requires a large number of samples for its Monte Carlo (MC) score estimate to have low variance and a large number of integration steps even when sampling from simple distributions. Also, its effectiveness highly depends on the choice of noise schedule and score clipping. These disadvantages demand careful hyperparameter tuning and raise issues when working with complicated energies.

---

*Equally contributed. Order is randomly assigned. Correspondence to `ro352@cam.ac.uk` and `bqiang@uw.edu`.

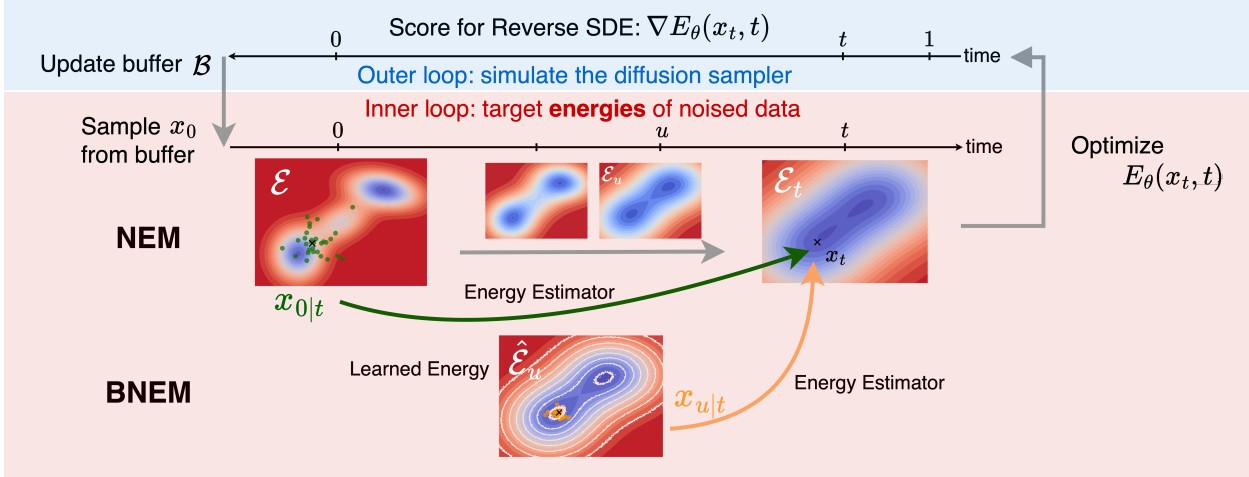

Figure 1: Both NEM and BNEM parameterize a time-dependent energy network $E_\theta(x_t, t)$ to target the energies of noised data. NEM targets an MC energy estimator computed from the target energy function; BNEM targets a Bootstrap energy estimator computed from learned energy functions at a slightly lower noise level. Contours are the ground truth energies at different noise levels; • represents samples used for computing the MC energy estimator, • represents samples used for computing the Bootstrap energy estimator, and the white contour line represents the learned energy at time $u$.

To further push the boundary of diffusion-based neural samplers, we propose NOISED ENERGY MATCHING (NEM), which learns a series of noised energy functions instead of the corresponding score functions. Despite a need to differentiate the energy network when simulating the diffusion sampler, NEM targets less noisy objectives as compared with iDEM. Additionally, using an energy-based parametrization enables NEM to use bootstrapping techniques for more efficient training and Metropolis-Hastings corrections for more accurate simulation. By applying the bootstrapping technique, we propose a variant of NEM called BOOTSTRAP NEM (BNEM). BNEM estimates high noise-level energies by bootstrapping from current energy estimates at slightly lower noise levels. BNEM increases bias but reduces variance in its training target.

Our methods outperform alternative baselines on both experiments with synthetic and $n$-body system targets. Additionally, we found that targeting energies instead of scores is more robust, requiring fewer MC samples during training and fewer integration steps during sampling. The latter compensates for the need of our methods to differentiate the energy networks during sampling for score calculation.

Our contributions are as follows:

- We introduce NEM and prove that targeting noised energies rather than noised scores has more advantages through a theoretical analysis.

- We apply bootstrapping energy estimation to NEM and present a theoretical analysis of the *Bias-Variance trade-off* for the resulting diffusion-based sampler.

- We perform experiments showing that BNEM and NEM significantly outperform alternative baseline neural samplers on four different tasks, while also demonstrating robustness under various hyper-parameter settings and efficiency in terms of number of function evaluations.

## 2  Preliminary

We consider learning a generative model for sampling from the Boltzmann distribution

$$\mu_{\text{target}}(x) = \frac{\exp(-\mathcal{E}(x))}{Z}, \quad Z = \int e^{-\mathcal{E}(x)} dx, \tag{1}$$

where $\mathcal{E}$ is the energy function and $Z$ is the intractable partition function. Efficiently sampling from this type of distribution is highly challenging. The recent success of Diffusion Models provides a promising way to tackle this problem.

**Diffusion Models (DMs)** (Sohl-Dickstein et al., 2015; Ho et al., 2020; Song et al., 2020) learn a generative process that starts from a known and tractable base distribution, *a.k.a.* denoising process, which is the inverse of a tractable noising process that starts from the target distribution. Formally, given samples from the target distribution, $x_0 \sim \mu_{\text{target}}$, the noising process is an SDE towards a known base distribution $p_1$, which is also known as the Diffusion SDE:

$$dx_t = f(x_t, t)dt + g(t)dw_t, \quad \text{for} \quad t \in [0, 1], \tag{2}$$

where $f(x_t, t)$ is called drift coefficient, $g(t)$ is the diffusion coefficient and $w_t$ is standard Brownian Motion. Diffusion Models work by approximately solving the following inverse SDE [1]:

$$dx_t = [f(x_t, t) - g^2(t)\nabla \log p_t(x_t)]dt + g(t)d\tilde{w}_t, \tag{3}$$

where $\tilde{w}_t$ is again standard Brownian Motion and $p_t$ is the marginal density of the diffusion SDE at time $t$. In the example of the Variance Exploding (VE) noising process, $f(x_t, t) \equiv 0$ and the perturbation kernel is given by $q_{t|0}(x_t|x_0) = \mathcal{N}(x_t; x_0, \sigma_t^2)$, where $\sigma_t^2 := \int g^2(s)ds$. Then the learning objective of DMs is obtained by using Tweedie's formula (Efron, 2011):

$$\mathcal{L}_{DM}(\theta) = \mathbb{E}_{p_{t|0}(x_t|x_0)p_0(x_0)p(t)} \left[ w(t) \left\| \frac{x_0 - x_t}{\sigma_t^2} - s_\theta(x_t, t) \right\|^2 \right], \tag{4}$$

where $w(t)$ is a positive weighting function and $s_\theta(x_t, t)$ is a score network, parameterized by $\theta$, which targets the conditional scores $\nabla \log p_{t|0}(x_t|x_0) = \nabla \log \mathcal{N}(x_t; x_0, \sigma_t^2 I)$. Minimising the above objective allows us to approximate the marginal scores $\nabla \log p_t(x_t)$ with $s_\theta(x_t, t)$.

## 3 Methods

We aim to train a diffusion-based neural sampler that samples from $\mu_{\text{target}}(x) = e^{-\mathcal{E}(x)}/Z$, where we assume we only have access to the energy function $\mathcal{E}$ without any known data from the target distribution, as illustrated in Fig. 1. To solve this problem, we introduce our proposed NEM framework and discuss the theoretical advantages of energy matching over score matching. Finally, we describe how bootstrapping (BNEM) can be used to obtain further gains. Full descriptions for training NEM and BNEM are provided in Algs. 1 and 2, and visualized in Fig. 1.

### 3.1 Denoising diffusion-based Boltzmann sampler

We consider training an energy-based diffusion sampler (Salimans & Ho, 2021; Gao et al., 2021) corresponding to a variance exploding (VE) noising process defined by $dx_t = g(t)dw_t$, where $t \in [0, 1]$, $g(t)$ is a function of time and $w_t$ is Brownian motion. The reverse SDE with Brownian motion $\bar{w}_t$ is $dx_t = -g^2(t)\nabla \log p_t(x_t)d_t + g(t)d\bar{w}_t$, where $p_t$ is the marginal of the diffusion process starting at $p_0 := \mu_{\text{target}}$.

Given the energy $\mathcal{E}(x)$ and the perturbation kernel $q_t(x_t|x_0) = \mathcal{N}(x_t; x_0, \sigma_t^2)$, where $\exp(-\mathcal{E}(x)) \propto p_0(x)$ and $\sigma_t^2 := \int_s^t g^2(s)ds$, one can obtain the marginal noised density $p_t$ as

$$p_t(x_t) \propto \int \exp(-\mathcal{E}(x_0))\mathcal{N}(x_t; x_0, \sigma_t^2 I)dx_0 = \mathbb{E}_{\mathcal{N}(x; x_t, \sigma_t^2 I)}[\exp(-\mathcal{E}(x))]. \tag{5}$$

Going a step further, the RHS of Eq. (5) defines a Boltzmann distribution over the noise-perturbed distribution $p_t$. The noised energy is defined as the negative logarithm of this unnormalized density

$$\mathcal{E}_t(x_t) := -\log \mathbb{E}_{\mathcal{N}(x; x_t, \sigma_t^2 I)}[\exp(-\mathcal{E}(x))], \quad \text{where} \quad \exp(-\mathcal{E}_t(x_t)) \propto p_t(x_t). \tag{6}$$

---

[1]For simplicity, we denote $\nabla_x$ by $\nabla$ throughout the paper.

**Training on MC estimated targets**   We can approximate the gradient of $\log p_t$ by fitting a score network $s_\theta(x_t, t)$ to the gradient of Monte Carlo (MC) estimates of Eq. (6), leading to iDEM (Akhound-Sadegh et al., 2024). The MC score estimator $S_K$ and the training objective are

$$S_K(x_t, t) := \nabla \log \frac{1}{K} \sum_{i=1}^{K} \exp(-\mathcal{E}(x_{0|t}^{(i)})), \quad x_{0|t}^{(i)} \sim \mathcal{N}(x; x_t, \sigma_t^2 I) \tag{7}$$

$$\mathcal{L}_{\text{DEM}}(x_t, t) := \|S_K(x_t, t) - s_\theta(x_t, t)\|^2. \tag{8}$$

Alternatively, we can fit an energy network $E_\theta(x_t, t)$ to MC estimates of Eq. (6). The gradient of this energy network w.r.t. input $x_t$, *i.e.* $\nabla E_\theta(x_t, t)$, can then be used to estimate the score required for diffusion-based sampling. The MC energy estimator $E_K$ and the training objective can be written as

$$E_K(x_t, t) := -\log \frac{1}{K} \sum_{i=1}^{K} \exp(-\mathcal{E}(x_{0|t}^{(i)})), \quad x_{0|t}^{(i)} \sim \mathcal{N}(x; x_t, \sigma_t^2 I) \tag{9}$$

$$\mathcal{L}_{\text{NEM}}(x_t, t) := \|E_K(x_t, t) - E_\theta(x_t, t)\|^2. \tag{10}$$

Notice that $S_K(x_t, t) = -\nabla E_K(x_t, t)$. To enable diffusion-based sampling, one is required to differentiate the energy network to obtain the marginal scores, *i.e.* $\nabla E_\theta(x_t, t)$, which doubles the computation of just evaluating $E_\theta(x_t, t)$. Regressing the MC energy estimator $E_K$ does not require to compute the gradient of the target energy $\mathcal{E}$ during training but it requires computing the gradient of $E_\theta$ during sampling. In other words, we change the requirement of having to differentiate the energy function $\mathcal{E}$ for training, to having to differentiate the learned neural network $E_\theta$ for sampling. Our method may have computational advantages when evaluating the gradient of the target energy is difficult or expensive.

**Bi-level iterative training scheme**   To train the diffusion model on the aforementioned Monte Carlo targets, we should choose the input locations $x_t$ at which the targets are computed. Previous works (Akhound-Sadegh et al., 2024; Midgley et al., 2023) used data points generated by a current learned denoising procedure. We follow their approach and use a bi-level iterative training scheme for noised energy matching. This involves

- An outer loop that simulates the diffusion sampling process to generate informative samples $x_0$. These samples are then used to update a replay buffer $\mathcal{B}$.

- A simulation-free inner loop that matches the noised energies (NEM) or scores (DEM) evaluated at noised versions $x_t$ of the samples $x_0$ stored in the replay buffer.

## 3.2   Energy-based learning v.s. score-based learning

In this section, we aim to answer the following questions.

(Q1) *Do NEM and iDEM differ with infinite data, training time, and model capacity?*

(Q2) *Why might NEM outperform iDEM in practical settings?*

To answer Q1, We assume that both NEM and iDEM use fully flexible energy and score networks and that both have converged by training with unlimited clean data. Since both networks are regressing on the MC estimates, at convergence, their network outputs are then the expectations $E_{\theta^*}(x_t, t) = \mathbb{E}[E_K(x_t, t)]$ and $s_{\theta^*}(x_t, t) = \mathbb{E}[S_K(x_t, t)]$. Noticing that $S_K = -\nabla E_K$ and by changing the order of differentiation and expectation, we show that NEM and iDEM result in the same score in this optimal learning setting, *i.e.* $s_{\theta^*} = -\nabla E_{\theta^*}$, in Appendix E.1.

However, factors such as limited data and model capacity, will always introduce learning error, which is related to the variance of targets Zhang et al. (2017); Belkin et al. (2019) and raises Q2. In the following, we characterize $E_K$ by quantifying its bias and variance in order to address this question.

**Proposition 3.1.** *If $\exp(-\mathcal{E}(x_{0|t}^{(i)}))$ is sub-Gaussian, then with probability $1 - \delta$ over $x_{0|t}^{(i)} \sim \mathcal{N}(x_t, \sigma_t^2)$, we have*

$$\|E_K(x_t, t) - \mathcal{E}_t(x_t)\| \leq \frac{\exp(\mathcal{E}_t(x_t))\sqrt{2v_{0t}(x_t)\log(2/\delta)}}{\sqrt{K}} + \mathcal{O}\left(\frac{1}{K}\right), \tag{11}$$

*where $v_{0t}(x_t) = \text{Var}_{\mathcal{N}(x;x_t,\sigma_t^2 I)}[\exp(-\mathcal{E}_t(x))]$.*

Proposition 3.1 shows an error bound on the training target used by NEM. This bound is invariant to energy shifts (i.e. adding a constant to the energy) because $\exp(\mathcal{E}_t(x_t))\sqrt{v_{0t}(x_t)}$ is invariant under that operation. The bound depends on energy values at $x_t$ and in the vecinity of $x_t$, *i.e.* $\mathcal{E}_t(x_t)$ and $v_{0t}(x_t)$. In particular, the error bound tends to be small in high-density regions (i.e., low-energy regions) where $\exp(\mathcal{E}_t(x_t))$ is relatively low. This yields reliable energy estimates that may be effectively leveraged for *Metropolis-Hastings corrections*. The bias and variance of $E_K$ can be characterised through the above error bound, as provided by the following corollary:

**Corollary 3.2.** *If $\exp(-\mathcal{E}(x_{0|t}^{(i)}))$ is sub-Gaussian, then the bias and variance of $E_K$ are given by*

$$\text{Bias}[E_K(x_t, t)] = \mathbb{E}[E_K(x_t, t)] - \mathcal{E}_t(x_t) = \frac{v_{0t}(x_t)}{2m_t^2(x_t)K} + \mathcal{O}\left(\frac{1}{K^2}\right), \tag{12}$$

$$\text{Var}[E_K(x_t, t)] = \frac{v_{0t}(x_t)}{m_t^2(x_t)K} + \mathcal{O}\left(\frac{1}{K^2}\right), \tag{13}$$

*where $m_t(x) = \exp(-\mathcal{E}_t(x_t))$ and $v_{0t}(x_t)$ is defined in Proposition 3.1.*

The complete proofs of Proposition 3.1 and Corollary 3.2 are given in Appendix C. Additionally, for 1-dimensional inputs, the variances of $S_K$ and $E_K$ can be related as follows.

**Proposition 3.3.** *Let $x \in \mathbb{R}$. If $\exp(-\mathcal{E}_t(x)) > 0$, $\exp(-\mathcal{E}(x_{0|t}^{(i)}))$ is sub-Gaussian, and $\|\nabla \exp(-\mathcal{E}(x_{0|t}^{(i)}))\|$ is bounded. Then*

$$\frac{Var[S_K(x_t, t)]}{Var[E_K(x_t, t)]} = 4(1 + \|\nabla \mathcal{E}_t(x_t)\|)^2 + \mathcal{O}\left(\frac{1}{K}\right). \tag{14}$$

Proposition 3.3 shows that the MC energy estimator has less variance than the MC score estimator in the 1-dimensional case, resulting in a less noisy training signal. The complete proof for the above results is provided in Appendix D. We argue that this variance gap naturally extends to higher-dimensional settings, which is trivial to show when there is perfect dependence or independence between data dimensions. Therefore, we claim that NEM provides an **easier learning problem** than iDEM. While NEM uses targets with lower variance, it still requires differentiating the energy $E_\theta$ during sampling, unlike iDEM, which directly uses the learned score $s_\theta$. This makes it hard to quantify the actual differences between these methods theoritically. Despite this, our experiments show that NEM performs better in practice than iDEM, in terms of both *performance* and *convergence rate*, indicating that the differentiation of $E_\theta$ during sampling does not introduce additional significant errors in NEM. To address potential memory issues arising from differentiating the energy network in high-dimensional tasks, we also introduce a memory-efficient sampling procedure in Appendix J.

### 3.3 Improvement with bootstrapped energy estimation

Using an energy network that directly models the noisy energy landscape has additional advantages. Intuitively, the variances of $E_K$ and $S_K$ explode at high noise levels as a result of the VE noising process. However, we can reduce the variance of the training target in NEM by using the learned noised energies at just slightly lower noise levels rather than using the target energy at time $t = 0$. Based on this, we propose Bootstrap

NEM, or **BNEM**, which uses a novel MC energy estimator at high noise levels that is bootstrapped from the learned energies at slightly lower noise levels. Suppose that $E_\theta(\cdot, s)$ is an energy network that already provides a relatively accurate estimate of the energy at a low noise level $s$, we can then construct a bootstrap energy estimator at a higher noise level $t > s$:

$$E_K(x_t, t, s; \theta) := -\log \frac{1}{K} \sum_{i=1}^K \exp(-E_\theta(x_{s|t}^{(i)}, s)), \quad x_{s|t}^{(i)} \sim \mathcal{N}\left(x; x_t, (\sigma_t^2 - \sigma_s^2)I\right), \tag{15}$$

$$\mathcal{L}_{\text{BNEM}}(x_t, t|s) := \|E_K(x_t, t, s; \text{SG}(\theta)) - E_\theta(x_t, t)\|^2, \tag{16}$$

where $\text{SG}(\theta)$ denotes a stop-gradient operation, preventing backpropagation through $\theta$.

Let us consider a bootstrapping-once scenario $(0 \to s \to t)$ where we assume that we first learn an optimal energy network at $s$ by using the MC energy estimator Eq. (9) with unlimited training data and a fully flexible model, resulting in $E_{\theta_s^*}(x_s, s) = \mathbb{E}[E_K(x_s, s)]$; then the Bootstrapped energy estimator is obtained by plugging $E_{\theta_s^*}$ into Eq. (15). Appendix G.1 shows that the bias of the bootstrapping-once estimator can then be characterized as

$$\text{Bias}[E_K(x_t, t, s; \theta_s^*)] = \frac{v_{0t}(x_t)}{2m_t^2(x_t)K^2} + \frac{v_{0s}(x_t)}{2m_s^2(x_t)K}.$$

This illustrates that, theoretically, bootstrapping once from time $s$ can quadratically reduce the bias contributed at time $t > s$, while the bias contributed at time $s$ is accumulated. By induction, given a chain of bootstrapping $0 = s_0 < ... < s_i < t$, we sequentially optimize the network at $(x_{s_j}, s_j)$ by bootstrapping from the previous level. Then the bias at $(x_t, t)$ can be charecterized as follows:

---

**Proposition 3.4.** *Given $\{s_i\}_{i=0}^n$ such that $\sigma_{s_i}^2 - \sigma_{s_{i-1}}^2 \le \kappa$, where $s_0 = 0$, $s_n = 1$ and $\kappa > 0$ is a small constant. suppose $E_\theta$ is incrementally optimized from $t \in [0,1]$ as follows: if $t \in [s_i, s_{i+1}]$, $E_\theta(x_t, t)$ targets an energy estimator bootstrapped from $\forall s \in [s_{i-1}, s_i]$ using Eq. (15). For $\forall 0 \le i \le n$ and $\forall s \in [s_{i-1}, s_i]$, the variance of the bootstrap energy estimator is approximately given by*

$$Var[E_K(x_t, t, s; \theta)] = \frac{v_{st}(x_t)}{v_{0t}(x_t)} Var[E_K(x_t, t)] \tag{17}$$

*and the bias of $E_K(x_t, t, s; \theta)$ is approximately given by*

$$\text{Bias}[E_K(x_t, t, s; \theta)] = \frac{v_{0t}(x_t)}{2m_t^2(x_t)K^{i+1}} + \sum_{j=1}^i \frac{v_{0s_j}(x_t)}{2m_{s_j}^2(x_t)K^j}, \tag{18}$$

*where $v_{yz}(x_z) = \text{Var}_{\mathcal{N}(x; x_z, (\sigma_z^2 - \sigma_y^2)I)}[\exp(-\mathcal{E}_y(x))]$ and $m_z(x_z) = \exp(-\mathcal{E}_z(x_z))$ for $\forall 0 \le y < z \le 1$.*

---

A detailed discussion and proof are given in Appendix G. Proposition 3.4 demonstrates that the bootstrap energy estimator, which estimates the noised energies by sampling from an $x_t$-mean Gaussian with smaller variance, can reduce the variance of the training target, because the variance ratio $v_{st}(x_t)/v_{0t}(x_t) < 1$ is very small, while this new target can introduce accumulated bias.

Proposition 3.4 shows that, the bias of BNEM consists of two components: (1) the target bias term which is reduced by a factor of $K^i$ compared to NEM, and (2) the sum of intermediate biases, which are each reduced by factors of $K^j$ where $j \ge 1$. Since $K$ is typically large and $i \ge 1$, the target bias term in BNEM is substantially smaller than in NEM. While BNEM does introduce additional accumulated terms, these terms are also reduced by powers of $K$, resulting in an overall lower bias compared to NEM under similar computational budgets. Therefore, with a proper choice of a number of MC samples $K$ and bootstrap trajectory $\{s_i\}_{i=1}^n$, the bias of BNEM (Eq. (12)) can be smaller than that of NEM (Eq. (18)) while enjoying a lower-variance training target.

**Variance-controlled bootstrap schedule** BNEM aims to trade the bias of the learning target to its variance. To ensure that the variance of the Bootstrap energy estimator at $t$ bootstrapped from $s$ is controlled

by a predefined bound $\beta$, *i.e.* $\sigma_t^2 - \sigma_s^2 \leq \beta$, we first split the time range $[0, 1]$ with $0 = t_0 < t_1 < ... < t_N = 1$ such that $\sigma_{t_{i+1}}^2 - \sigma_{t_i}^2 \leq \beta/2$; then we uniformly sample $s$ and $t$ from adjacent time splits during training for variance control.

**Training of BNEM** To train BNEM, it is crucial to account for the fact that bootstrap energy estimation at $t$ can only be accurate when the noised energy at $s$ is well-learned. Given a partially trained NEM energy network, we introduce a rejection scheme for bootstrap fine-tuning to balance the bootstrapped and original MC estimators. Specifically, at each training step, we compute the normalized losses $\mathcal{L}_{\text{NEM}}(x_t, t)/\sigma_t^2$ and $\mathcal{L}_{\text{NEM}}(x_s, s)/\sigma_s^2$, and define an acceptance ratio proportional to their values. If accepted, we use the bootstrapped estimator; otherwise, we fall back to the original MC estimator. The detailed illustration could be found in Appendix B.2 and Alg. 2.

## 4 Related works

**Flow-based Neural Sampler.** Flow AIS Bootstrap (Midgley et al., 2023, FAB) trains Normalizing Flows to minimize the $\alpha$=2-divergence, *i.e.* the variance of importance weights, which exhibits mode covering properties. Notably, it employs a replay buffer for better exploitation.

**Path-measure-based Neural Sampler.** One can generate samples by a sequence of actions, which defines a path of the sampling process. Path-measure-based methods train neural samplers by minimizing the divergence between the sampling path and its reversal. Inspired by diffusion models (Song & Ermon, 2019; Ho et al., 2020), Zhang & Chen (2022); Vargas et al. (2023a); Richter & Berner (2023) propose to learn the sampling process by matching the time reversal of the diffusion process. Doucet et al. (2022) proposes to learn the time-reversal of the AIS path, while Vargas et al. (2023c); Albergo & Vanden-Eijnden (2024) introduce a learnable corrector to make sure the sampler transports between a sequence of predefined marginal densities. On the other hand, inspired by reinforcement learning, Bengio et al. (2021) proposes GFlowNets, which amortize MCMC by utilizing local information, such as (sub-)trajectory detailed-balance, to train a generative model. Sendera et al. (2024) proposes techniques to improve GFlowNets. However, most of these methods require simulation during training, which still poses challenges for scaling up to higher-dimensional tasks.

**Monte-Carlo-based Neural Sampler.** Huang et al. (2023) proposes to estimate the denoising score using a Monte Carlo method. To boost scalability, iDEM (Akhound-Sadegh et al., 2024) introduces an off-policy training loop to regress an MC score estimator. It achieves previous state-of-the-art performance on the tasks below. Notably, this MC estimator is used in Vargas et al. (2023b); Grenioux et al. (2024) and resembles an importance-weighted estimator of the Target Score Identity proposed by Bortoli et al. (2024). Woo & Ahn (2024) proposes a variant to target the vector field.

**Diffusion-based MCMC Sampler.** Chen et al. (2024); Grenioux et al. (2024) combine MCMC with diffusion sampling, by a Gibbs-style sampling procedure which alternates between the clean data space and the noisy data space.

**Recent developments.** Since the initial submission of this work in early 2025, the field of neural samplers has experienced rapid progress. To account for these recent advancements without disrupting the core narrative of our original study, we provide an extended discussion of contemporaneous literature in Appendix A.

## 5 Experiments

We evaluate our methods and baseline models on 4 potentials. A complete description of all energy functions, metrics, and experiment setups is in Appendix L. Supplementary experiments can be found in Appendix M. We provide an anonymous link to our code for implementation reference[2].

**Datasets.** We evaluate all neural samplers on 4 different problems: a GMM with 40 modes ($d = 2$), a 4-particle double-well (DW-4) potential ($d = 8$), a 13-particle Lennard-Jones (LJ-13) potential ($d = 39$) and a 55-particle Lennard-Jones (LJ-55) potential ($d = 165$). For LJ-n potentials, the energy can be extreme

---

[2]https://anonymous.4open.science/r/BNEM-NeurIPS2025/README.md

Table 1: Performance comparison of neural samplers across four target distributions. We evaluate each method using three metrics: data Wasserstein-2 distance (x-$\mathcal{W}_2$), energy Wasserstein-2 distance ($\mathcal{E}$-$\mathcal{W}_2$), and Total Variation (TV). Each sampler is run with three random seeds; we report the mean $\pm$ standard deviation. $*$ indicates divergent training. $\dagger$ denotes cases where only one seed converged, with the best result reported. Best values are in **bold**.

| Target → | **GMM-40** ($d=2$) | | | **DW-4** ($d=8$) | | | **LJ-13** ($d=39$) | | | **LJ-55** ($d=165$) | | |
|---|---|---|---|---|---|---|---|---|---|---|---|---|
| Sampler ↓ | x-$\mathcal{W}_2$ | $\mathcal{E}$-$\mathcal{W}_2$ | TV | x-$\mathcal{W}_2$ | $\mathcal{E}$-$\mathcal{W}_2$ | TV | x-$\mathcal{W}_2$ | $\mathcal{E}$-$\mathcal{W}_2$ | TV | x-$\mathcal{W}_2$ | $\mathcal{E}$-$\mathcal{W}_2$ | TV |
| DDS | $15.04_{\pm2.97}$ | $305.13_{\pm186.06}$ | $0.96_{\pm0.01}$ | $0.82_{\pm0.21}$ | $558.79_{\pm787.92}$ | $0.38_{\pm0.14}$ | $*$ | $*$ | $*$ | $*$ | $*$ | $*$ |
| PIS | $6.58_{\pm1.68}$ | $79.86_{\pm7.79}$ | $0.95_{\pm0.01}$ | $*$ | $*$ | $*$ | $*$ | $*$ | $*$ | $*$ | $*$ | $*$ |
| FAB | $9.08_{\pm1.41}$ | $47.60_{\pm7.24}$ | $0.79_{\pm0.07}$ | $0.62_{\pm0.02}$ | $112.70_{\pm20.33}$ | $0.38_{\pm0.02}$ | $*$ | $*$ | $*$ | $*$ | $*$ | $*$ |
| iDEM | $8.21_{\pm5.43}$ | $60.49_{\pm70.12}$ | $0.82_{\pm0.03}$ | $0.50_{\pm0.03}$ | $2.80_{\pm1.72}$ | $0.16_{\pm0.01}$ | $0.87_{\pm0.00}$ | $6770\dagger$ | $0.06_{\pm0.01}$ | $1.98_{\pm0.01}$ | $12615.77_{\pm991.43}$ | $0.14_{\pm0.01}$ |
| NEM | $5.28_{\pm0.89}$ | $44.56_{\pm39.56}$ | $0.91_{\pm0.02}$ | $\mathbf{0.48}_{\pm0.02}$ | $0.85_{\pm0.52}$ | $\mathbf{0.14}_{\pm0.01}$ | $0.87_{\pm0.01}$ | $5.01_{\pm3.14}$ | $\mathbf{0.03}_{\pm0.00}$ | $1.90_{\pm0.01}$ | $\mathbf{118.58}_{\pm106.63}$ | $0.10_{\pm0.02}$ |
| BNEM | $\mathbf{3.66}_{\pm0.30}$ | $\mathbf{1.87}_{\pm1.00}$ | $\mathbf{0.79}_{\pm0.04}$ | $0.49_{\pm0.02}$ | $\mathbf{0.38}_{\pm0.09}$ | $\mathbf{0.14}_{\pm0.01}$ | $\mathbf{0.86}_{\pm0.00}$ | $\mathbf{1.02}_{\pm0.69}$ | $\mathbf{0.03}_{\pm0.00}$ | $\mathbf{1.88}_{\pm0.01}$ | $119.46_{\pm77.92}$ | $\mathbf{0.08}_{\pm0.01}$ |

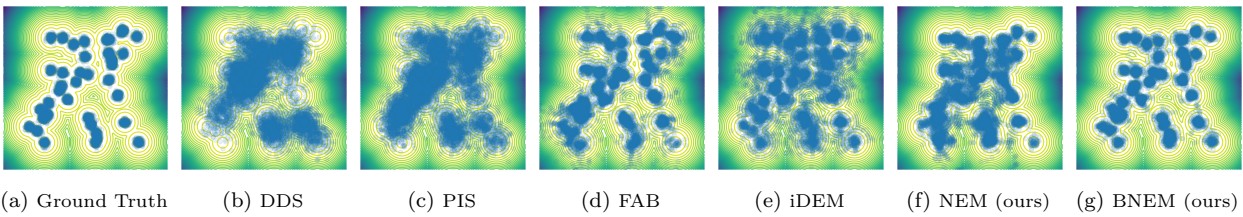

(a) Ground Truth  (b) DDS  (c) PIS  (d) FAB  (e) iDEM  (f) NEM (ours)  (g) BNEM (ours)

Figure 2: Sampled points from samplers applied to GMM-40 potentials, with the ground truth represented by contour lines. For diffusion-based methods, the reverse SDE integration steps are limited to 100.

when particles are too close to each other, creating problems for estimating noised energies. To overcome this issue, we smooth the Lennard-Jones potential through the cubic spline interpolation, according to Moore et al. (2024).

**Baseline.** We compare NEM and BNEM to the following recent works: Denoising Diffusion Sampler (DDS, Vargas et al., 2023a), Path Integral Sampler (PIS, Zhang & Chen, 2022)), Flow Annealed Bootstrap (FAB, Midgley et al., 2023) and Iterated Denoising Energy Matching (iDEM, Akhound-Sadegh et al., 2024). Due to the high complexity of DDS and PIS training due to their simulation-based nature, we limit their integration step when sampling to 100. As shown in prior work (Akhound-Sadegh et al., 2024), all baselines perform well on the GMM benchmark when using 1000 integration steps—our methods as well. To increase task difficulty and better assess robustness, we reduce both the number of integration steps and MC samples to 100 in our evaluation. For other tasks, *i.e.* DW-4, LJ-13 and LJ-55, we stick to using 1000 steps for reverse SDE integration and 1000 MC samples in the estimators. We notice that the latest codebase of iDEM uses 100 MC samples but employs 10 steps of Langevin dynamics before evaluation in the LJ-55 task. For fair comparison, we disable these Langevin steps [3] and stick to using 1000 MC samples for iDEM. For BNEM, we set $\beta = 0.1$ for all tasks. All samplers are trained using an NVIDIA-A100 GPU.

**Architecture.** We use the same network architecture (MLP for GMM and EGNN for $n$-body systems, *i.e.* DW-4, LJ-13, and LJ-55) in DDS, PIS, iDEM, NEM and BNEM. To ensure a similar number of parameters for each sampler, if the score network is parameterized by $s_\theta(x,t) = f_\theta(x,t)$, the energy network is set to be $E_\theta(x,t) = \mathbf{1}^\top f_\theta(x,t) + c$ with a learnable scalar $c$. Furthermore, this setting ensures SE(3)-invariance for the energy network. Since FAB requires an invertible architecture, we use in this method a continuous normalizing flow specified by $f_\theta(x,t)$ so that it has a similar number of parameters as the other samplers.

**Metrics.** We applied the 2-Wasserstein distances on data (x-$\mathcal{W}_2$) and energy ($\mathcal{E}$-$\mathcal{W}_2$) values as quality metrics. Additionally, the Total Variation (TV) is also computed. TV is computed directly from data in GMM experiments and from interatomic distances in $n$-body systems. To compute $\mathcal{W}_2$ and TV metrics, we use pre-generated ground-truth samples: (a) For GMM, we sample from the ground truth distribution; (b) For DW-4, LJ-13 and LJ-55, we use samples from Klein et al. (2023b). Note that we take SE(3) into account

---

[3] In fact, running any MCMC on top of the generated samples is trivial, which can be employed by **any** of the baselines.

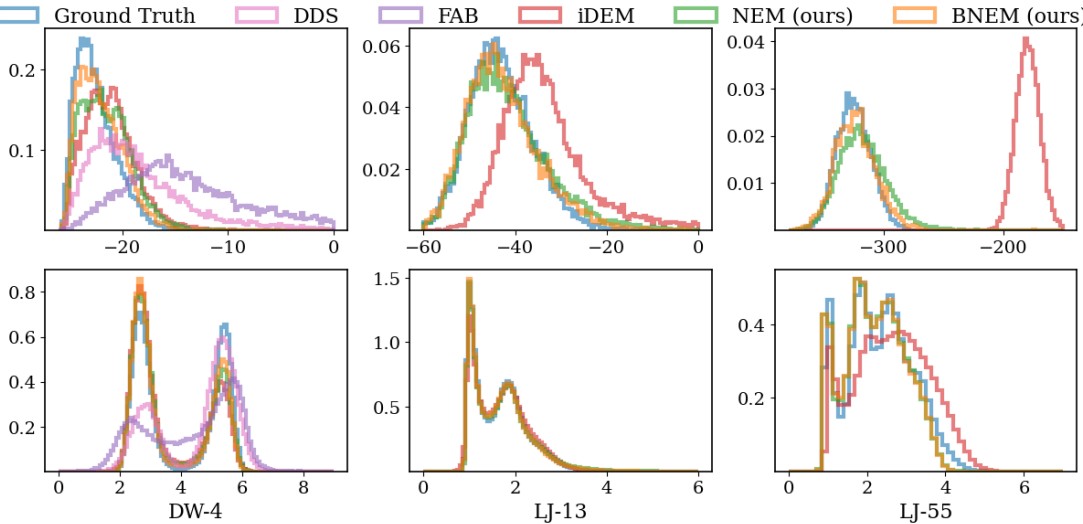

Figure 3: Histogram for energy (**top**) and interatomic distance (**bottom**) of generated samples.

when calculating $\mathbf{x}$-$\mathcal{W}_2$ for $n$-body systems . For a detailed description of the evaluation metrics, please refer to Appendix L.2.

## 5.1 Main Results

We report x-$\mathcal{W}_2$, $\mathcal{E}$-$\mathcal{W}_2$, and TV for all tasks in Table 1. The table demonstrates that by targeting less noisy objectives, NEM outperforms DEM on most metrics, particularly for complex tasks such as LJ-13 and LJ-55. Fig. 2 visualizes the generated samples from each sampler in the GMM benchmark. When the compute budget is constrained—by reducing neural network size for DDS, PIS, and FAB, and limiting the number of integration steps and MC samples to 100 for all baselines—none of them achieve high-quality samples with sufficient mode coverage. In particular, iDEM produces samples that are not concentrated around the modes in this setting. Conversely, NEM generates samples with far fewer outliers and achieves the best performance on all metrics. BNEM can further improve on top of NEM, generating data that are most similar to the ground truth ones.

For the equivariant tasks, *i.e.* DW-4, LJ-13, and LJ-55, we compute the energies and interatomic distances for each generated sample. Empirically, we found that smoothing the energy-distance function by a cubic spline interpolation (Moore et al., 2024) to avoid extreme values (i.e. when the particles are too close) is a key step when working with the Lennard Jones potential. Furthermore, this smoothing technique can be applied in a wide range of many-particle systems (Pappu et al., 1998). Therefore, NEM and BNEM can be applied without significant modeling challenges. We also provide an ablation study on applying this energy-smoothing technique to score-based iDEM. The results in Table 3 suggest that it could help to improve the performance of iDEM but NEM still outperforms it.

Fig. 3 shows the histograms of the energies and interatomic distances on each $n$-particle system. It shows that both NEM and BNEM closely match the ground truth densities, outperforming all other baselines. Moreover, for the most complex task, LJ-55, NEM demonstrates greater stability during training, generating more low-energy samples, unlike iDEM, which is more susceptible to instability and variance from different random seeds.

$\mathcal{E}$-$\mathcal{W}_2$ is susceptible to outliers with high energy, especially in complex tasks like LJ-13 and LJ-55, where an outlier that corresponds to a pair of particles that are close to each other can result in an extremely large value of this metric. We find that for LJ-n tasks, NEM and BNEM tend to generate samples with low energies and result in low $\mathcal{E}$-$\mathcal{W}_2$, while iDEM can produce high energy outliers and therefore results in the extremely high values of this metric. We also notice that for LJ-55, the mean value of $\mathcal{E}$-$\mathcal{W}_2$ of BNEM is similar to NEM, while its results have smaller variance indicating a better result. Overall, NEM consistently

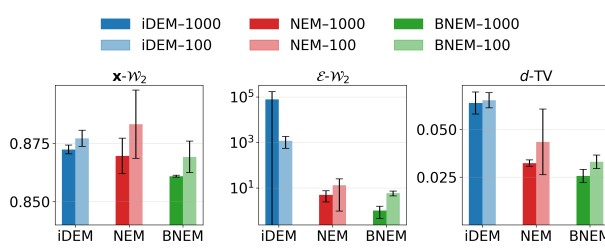

Figure 4: Barplots comparing iDEM, NEM, and BNEM evaluations with 1000 vs. 100 integration steps and MC samples on the LJ-13 benchmark.

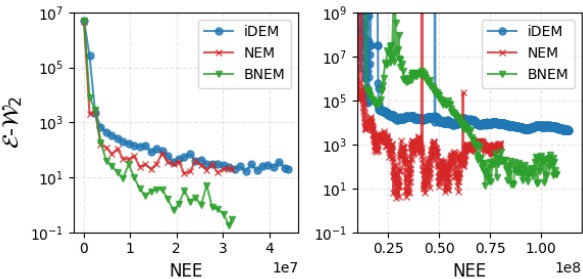

Figure 5: Number of energy evaluations vs. $\mathcal{E}$-$\mathcal{W}_2$. Left: GMM; Right: LJ-55. The y-axis is on a logarithmic scale.

outperforms iDEM, while BNEM can further improve over NEM and achieve state-of-the-art performance in all tasks.

**Robustness of NEM and BNEM.** Table 1 highlights the superior robustness of NEM and BNEM, requiring fewer integration steps and MC samples compared to iDEM and other baselines. We further examine this robustness using the LJ-13 benchmark to showcase the advantages of NEM and BNEM. Fig. 4 illustrates how metrics change when reducing integration steps and MC samples from 1000 to 100. A comprehensive comparison across different computational budgets is provided in Table 4 (Appendix M). Results indicate that under constrained budgets, iDEM performance significantly deteriorates in GMM and DW-4 potentials, whereas NEM remains more robust. In LJ-13, both methods experience degradation, yet NEM-100 surpasses iDEM-100 and even iDEM-1000 in terms of $\mathcal{E}$-$\mathcal{W}_2$ and TV metrics. Moreover, BNEM-100 maintains strong performance, matching iDEM-1000 in GMM and DW-4, and notably outperforming iDEM-1000 in the LJ-13 benchmark.

**Efficiency of NEM and BNEM.** Fig. 5 compares the efficiency of NEM and BNEM against iDEM. On the simple GMM task, NEM reaches the same optimality as iDEM but requires 5–10 times fewer energy evaluations. BNEM achieves similar results an order of magnitude faster and delivers better performance. On the more challenging LJ55 task, all three methods require similar energy evaluations for convergence. However, NEM and BNEM yield an order of magnitude improvement over iDEM. Notably, BNEM converges more slowly than NEM due to bootstrapping instability but eventually reaches a more stable optimum with significantly lower variance in $\mathcal{E}$-$\mathcal{W}_2$. These results align with the variance advantage of energy estimators over score estimators established in Proposition 3.3, suggesting that the lower-variance MC energy estimator offers a more stable training signal. We report the memory and runtime (per-loop) overhead of iDEM, NEM, and BNEM in Tables 7 and 8; see Appendix M.5 for more details.

## 6 Conclusion

We have proposed NEM and BNEM, two neural samplers targeting Boltzmann distributions. NEM regresses a novel Monte Carlo energy estimator with reduced bias and variance, achieving faster convergence rate and better optimality with more robustness to hyper-parameters when compared to iDEM. BNEM builds on NEM, employing an energy estimator bootstrapped from lower noise-level data, theoretically trading bias for variance. Empirically, NEM outperforms baselines on 4 different benchmarks, and BNEM improves over NEM .

**Limitations and future work.** Though NEM/BNEM achieve better performance on multiple benchmark sampling tasks, they have to differentiate through a neural network, which can increase memory cost. To mitigate this, we have explored using an alternative estimator to reduce the memory trace for energy-based modeling in Appendices J and M.7. Another limitation is that, to stabilize training with extreme energy values, we had to use a cubic-spline interpolation technique for the LJ-n potentials, which can introduce small

biases in the generated samples. Future work could avoid this by using contrastive learning or an alternative energy training scheme.

## Broader Impact

This paper presents work whose goal is to advance the field of Machine Learning. There are many potential societal consequences of our work, none which we feel must be specifically highlighted here.

## Acknowledgements

We acknowledge Javier Antorán, Wenlin Chen, Louis Grenioux, Jiajun He, Zixing Song, and Yusong Wang (listed alphabetically) for helpful and inspiring discussions. JMHL and RKOY acknowledge the support of a Turing AI Fellowship under grant EP/V023756/1. This project acknowledges the resources provided by the Cambridge Service for Data-Driven Discovery (CSD3) operated by the University of Cambridge Research Computing Service (www.csd3.cam.ac.uk), provided by Dell EMC and Intel using Tier-2 funding from the Engineering and Physical Sciences Research Council (capital grant EP/T022159/1), and DiRAC funding from the Science and Technology Facilities Council (www.dirac.ac.uk).

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

# BNEM: A Boltzmann Sampler Based on Bootstrapped Noised Energy Matching Appendix

## A Extended Related Works

In this section, we discuss recent advancements in the fields of neural samplers and energy-based models that have emerged concurrently with this work.

**Path-Measure-Based Neural Samplers.** Recent methods have focused on improving neural samplers using path measures induced by diffusion processes. Havens et al. (2025) proposes *Adjoint Sampling* (AS), a scalable improvement over Akhound-Sadegh et al. (2024) derived from a *Stochastic Optimal Control* (SOC) perspective. Liu et al. (2025) extends AS to a Schrödinger Bridge setting (ASBS). Furthermore, Blessing et al. (2026) unifies AS and ASBS from a probabilistic standpoint and proposes a variant that yields additional performance boosts.

**Enhancing Neural Samplers via Annealing.** Traditionally, neural samplers have been trained using a fixed temperature. However, recent progress has highlighted the benefits of annealing techniques, which begin with an easier-to-sample high-temperature distribution and progressively anneal to a lower-temperature one. Rissanen et al. (2025); Akhound-Sadegh et al. (2024) first introduced the integration of annealing into neural sampler training. Building on this, Blessing et al. (2025) systematically analyzes the annealing process through a trust-region perspective, where the annealing acts as a constraint on the KL divergence between the previous and updated samplers. This approach provides an automated and optimal temperature schedule, eliminating the need for manual tuning.

**Energy-Based Training.** Both NEM and BNEM primarily regress the marginal energy defined by the diffusion process. However, instead of relying solely on the energy estimators (Equations (9) and (15)), one can leverage a replay buffer to jointly optimize data-driven energy-based losses. Recent works (Aggarwal et al., 2025; He et al., 2026; OuYang et al., 2026) propose various regularizers for training energy-based diffusion models, which could potentially be adapted to enhance the training of NEM/BNEM. Intuitively, these regularizers encourage greater and more efficient exploitation of the replay buffer.

## B Details of Training NEM & BNEM

### B.1 Iterated Training for NEM and BNEM

### B.2 Inner-loop of BNEM Training

In the training of BNEM, we first use NEM to obtain an initial energy network and then apply training with the bootstrapped estimator. We then aim to use bootstrapping only when the energy network is significantly better at time $s$ than at time $t$. We quantify this by evaluating the NEM losses at times $t$ and $s$ given a clean data point $x_0$. In particular, we could compare $\tilde{l}_s(x_s) = \mathcal{L}_{\text{NEM}}(x_s, s)$ and $\tilde{l}_t(x_t) = \mathcal{L}_{\text{NEM}}(x_t, t)$, where $x_s \sim \mathcal{N}(x_0, \sigma_s^2 I)$ and $x_t \sim \mathcal{N}(x_0, \sigma_t^2 I)$. However, a direct comparison of $\tilde{l}_s$ and $\tilde{l}_t$ is not reliable because the intrinsic difficulty of the regression problem changes with time: even for an *optimal* regressor under squared loss, the minimum achievable MSE equals the conditional variance of the regression target, i.e., $\inf_f \mathbb{E}\|Y - f(X)\|^2 = \mathbb{E}[\text{Var}(Y \mid X)]$. Along the noising path in Diffusion models, this variance typically

---

**Algorithm 1** Iterated training for (Bootstrapped) Noised Energy Matching

---

**Input:** Energy network $E_\theta$, Batch size $b$, Noise schedule $\sigma_t^2$, Replay buffer $\mathcal{B}$, Num. MC samples $K$, $L$-step Diffusion solver $D_L$

1: **while** Outer-Loop **do**
2:     $\mathcal{B} = \text{Update}(\mathcal{B}, D_L(-\nabla E_\theta, b))$
3:     **while** Inner-Loop **do**
4:         $x_0 \sim \mathcal{B}, t \sim \mathcal{U}(0,1), x_t \sim \mathcal{N}(x_0, \sigma_t^2)$
5:         $\mathcal{L}_{\text{NEM}}(x_t, t) = \|E_K(x_t, t) - E_\theta(x_t, t)\|^2$
6:         $\mathcal{L}_{\text{BNEM}}(x_t, t) = \|E_K(x_t, t, s, \text{SG}(\theta)) - E_\theta(x_t, t)\|^2, s \sim \mathcal{U}(t_i, t)$ where $t_i \le t < t_{i+1}$
7:         $\theta \leftarrow \text{Update}(\theta, \nabla_\theta \mathcal{L}_{\text{NEM}})$
8:     **end while**
9: **end while**

---

increases with the noise level, so a smaller raw loss at time $s$ does not necessarily imply that the energy network is better learned at $s$ than at $t$. Heuristically, the conditional variance of the NEM regression target scales with the variance of the noising kernel, which is $\sigma_t^2$. To factor out this time-dependent scale and compare losses in a roughly "noise-normalized" unit, we instead use $l_s(x_s) = \mathcal{L}_{\text{NEM}}(x_s, s)/\sigma_s^2$ and $l_t(x_t) = \mathcal{L}_{\text{NEM}}(x_t, t)/\sigma_t^2$. We then apply a rejection-based training rule—rather than a hard switch—to smooth this noisy loss ratio and improve training stability, as follows.

(a) Given $s$, $t$ and $x_0$, we first noise $x_0$ to $s$ and $t$ respectively, and compute the normalized losses defined above, *i.e.* $l_s(x_s)$ and $l_t(x_t)$;

(b) These losses indicate how well the energy network fits the noised energies at different times; we then compute $\alpha = \min(1, l_s(x_s)/l_t(x_t))$;

(c) With probability $\alpha$, we accept targeting an energy estimator at $t$ bootstrapped from $s$ and otherwise, we stick to targeting the original MC energy estimator.

---

**Algorithm 2** Inner-loop of Bootstrap Noised Energy Matching training

---

**Input:** Network $E_\theta$, Batch size $b$, Noise schedule $\sigma_t^2$, Replay buffer $\mathcal{B}$, Num. MC samples $K$

1: **while** Inner-Loop **do**
2:     $x_0 \leftarrow \mathcal{B}.\text{sample}()$
3:     $t \sim \mathcal{U}(0,1), x_t \sim \mathcal{N}(x_0, \sigma_t^2)$
4:     $n \leftarrow \arg\{i : t \in [t_i, t_{i+1}]\}$
5:     $s \sim \mathcal{U}(t_{n-1}, t_n), x_s \sim \mathcal{N}(x_0, \sigma_s^2)$
6:     $l_s(x_s) \leftarrow \|E_K(x_s, s) - E_\theta(x_s, s)\|^2/\sigma_s^2$
7:     $l_t(x_t) \leftarrow \|E_K(x_t, t) - E_\theta(x_t, t)\|^2/\sigma_t^2$
8:     $\alpha \leftarrow \min(1, l_t(x_t)/l_s(x_s))$
9:     with probability $\alpha$,
10:         $\mathcal{L}_{\text{BNEM}}(x_t, t) = \mathcal{L}_{\text{BNEM}}(x_t, t|s)$
11:     Otherwise,
12:         $\mathcal{L}_{\text{BNEM}}(x_t, t) = \mathcal{L}_{\text{NEM}}(x_t, t)$
13:     $\theta \leftarrow \text{Update}(\theta, \nabla_\theta \mathcal{L}_{\text{BNEM}})$
14: **end while**

---

## C   Proof of Proposition 3.1 and Corollary 3.2

---

**Proposition 3.1.** *If $\exp(-\mathcal{E}(x_{0|t}^{(i)}))$ is sub-Gaussian, then with probability $1-\delta$ over $x_{0|t}^{(i)} \sim \mathcal{N}(x_t, \sigma_t^2)$, we have*

$$\|E_K(x_t, t) - \mathcal{E}_t(x_t)\| \leq \frac{\sqrt{2 v_{0t}(x_t)} \exp(\mathcal{E}_t(x_t)) \sqrt{\log(2/\delta)}}{\sqrt{K}} + \mathcal{O}\left(\frac{1}{K}\right) \tag{19}$$

*where $v_{0t}(x_t) = \mathrm{Var}_{\mathcal{N}(x;x_t,\sigma_t^2 I)}[\exp(-\mathcal{E}_t(x))]$.*

---

**Corollary 3.2.** *If $\exp(-\mathcal{E}(x_{0|t}^{(i)}))$ is sub-Gaussian, then the bias and variance of $E_K$ are given by*

$$\mathrm{Bias}[E_K(x_t, t)] = \mathbb{E}[E_K(x_t, t)] - \mathcal{E}_t(x_t) = \frac{v_{0t}(x_t)}{2 m_t^2(x_t) K} + \mathcal{O}\left(\frac{1}{K^2}\right), \tag{20}$$

$$\mathrm{Var}[E_K(x_t, t)] = \frac{v_{0t}(x_t)}{m_t^2(x_t) K} + \mathcal{O}\left(\frac{1}{K^2}\right), \tag{21}$$

*where $m_t(x) = \exp(-\mathcal{E}_t(x_t))$ and $v_{0t}(x_t)$ is defined in Proposition 3.1.*

---

**Proof.**   Let's define the following variables

$$m_t(x_t) = \exp(-\mathcal{E}_t(x_t)) \tag{22}$$

$$v_{st}(x_t) = \mathrm{Var}_{\mathcal{N}(x_t,(\sigma_t^2 - \sigma_s^2)I)}[\exp(-\mathcal{E}(x))] \tag{23}$$

Notice that $E_K$ is a logarithm of an unbiased estimator and by the sub-Gaussian assumption of $\exp(-\mathcal{E}(x_{0|t}^{(i)}))$, one can derive that $E_K$ is also sub-Gaussian. Furthermore, the mean and variance can be approximated using a Taylor expansion, a technique formally known as the Delta Method Casella & Berger (2002). Specifically, if a random variable X has mean m and variance v, then its logarithm logX has the following mean and variance:

$$\mathbb{E}[\log X] = \log m - \frac{v}{2m^2} + \mathcal{O}(v^2), \quad \mathrm{Var}(\log X) = \frac{v}{m^2} + \mathcal{O}(v^2). \tag{24}$$

Hence, according to the sub-Gaussianess of $E_K$'s negative exponential, we have

$$\mathbb{E}E_K(x_t, t) = \mathcal{E}_t(x_t) + \frac{v_{0t}(x_t)}{2 m_t^2(x_t) K} + \mathcal{O}\left(\frac{1}{K^2}\right) \tag{25}$$

$$\mathrm{Var}[E_K(x_t, t)] = \frac{v_{0t}(x_t)}{m_t^2(x_t) K} + \mathcal{O}\left(\frac{1}{K^2}\right) \tag{26}$$

Recall that a random variable $X$ is $\sigma$-sub-Gaussian if

$$P(\|X - \mathbb{E}X\| \geq \epsilon) \leq 2 \exp\left(-\frac{\epsilon^2}{2\sigma^2}\right), \tag{27}$$

for any $\epsilon > 0$. Equivalently, for any $\delta \in (0, 1)$, with probability at least $1 - \delta$, we have

$$\|X - \mathbb{E}X\| \leq \sqrt{2\sigma^2 \log \frac{2}{\delta}}. \tag{28}$$

Hence, one can obtain the concentration inequality for $E_K$ by incorporating the sub-Gaussianess. For any $\delta \in (0, 1)$, with probability $1 - \delta$, we have

$$\|E_K(x_t, t) - \mathbb{E}[E_K(x_t, t)]\| \leq \sqrt{2 \frac{v_{0t}(x_t)}{m_{0t}^2(x_t) K} \log \frac{2}{\delta}} \tag{29}$$

By using the above Inequality Eq. (29) and the triangle inequality

$$\|E_K(x_t, t) - \mathcal{E}_t(x_t)\| \leq \|E_K(x_t, t) - \mathbb{E}[E_K(x_t, t)]\| + \|\mathbb{E}[E_K(x_t, t)] - \mathcal{E}_t(x_t)\| \tag{30}$$

$$= \|E_K(x_t, t) - \mathbb{E}[E_K(x_t, t)]\| + \frac{v_{0t}(x_t)}{2m_t^2(x_t)K} + \mathcal{O}(1/K^2) \tag{31}$$

$$\leq \sqrt{2\frac{v_{0t}(x_t)}{m_t^2(x_t)K} \log\frac{2}{\delta}} + \frac{v_{0t}(x_t)}{2m_t^2(x_t)K} + \mathcal{O}(1/K^2) \tag{32}$$

$$= \frac{\sqrt{2v_{0t}(x_t)}\exp(\mathcal{E}_t(x_t))\sqrt{\log(2/\delta)}}{\sqrt{K}} + \mathcal{O}(1/K) \tag{33}$$

holds with probability $1 - \delta$ for any $\delta \in (0, 1)$.

## C.1   $E_K$ error bound v.s. $S_K$ error bound

To connect the bias of MC energy estimator and the MC score estimator, we introduce the error bound of the MC score estimator $S_K$, where $S_K = \nabla E_K$, proposed by Akhound-Sadegh et al. (2024) as follows

$$\|S_K(x_t, t) - S(x_t, t)\| \leq \frac{2C\sqrt{\log(\frac{2}{\delta})}(1 + \|\nabla \mathcal{E}_t(x_t)\|)\exp(\mathcal{E}_t(x_t))}{\sqrt{K}} \tag{34}$$

which also assumes that $\exp(-\mathcal{E}(x_{0|t}^{(i)}))$ is sub-Gaussian and further assumes sub-Gaussianess over $\|\nabla \mathcal{E}(x_{0|t}^{(i)})\|$. On the other hand, by the sub-Gaussianess assumption of $\exp(-\mathcal{E}(x_{0|t}^{(i)}))$, it's easy to show that the constant term $C$ in Equation 34 is $C = \sqrt{2v_{0t}(x_t)}$.

Therefore, we show that the energy estimator error bound would always be $2(1 + \|\nabla \mathcal{E}_t(x_t)\|)$ times tighter than the score estimator error bound.

## C.2   Invariant of the error bounds

At the end, we show that the above error bounds for both $E_K$ and $S_K$ is invariant to any energy shift, *i.e.* a shifted system energy $\mathcal{E}(x) + C$ with any real value constant $C$ will always result in the same bounds.

The proof is simple. By noticing that

$$\tilde{v}_{st}(x_t) = \mathrm{Var}_{\mathcal{N}(x_t, (\sigma_t^2 - \sigma_s^2)I)}[\exp(-\mathcal{E}(x) + C)] \tag{35}$$

$$= e^{-2C}\mathrm{Var}_{\mathcal{N}(x_t, (\sigma_t^2 - \sigma_s^2)I)}[\exp(-\mathcal{E}(x))] \tag{36}$$

$$\tilde{m}_t(x_t) = \int \exp(-\mathcal{E}(x + C))\mathcal{N}(x; x_t, \sigma_t^2 I) \tag{37}$$

$$= e^{-C}m_t(x_t) \tag{38}$$

the exponential shifted value will be cancelled when calculating $v_{st}(x_t)/m_t^2(x_t)$. $\qquad\square$

## D   Proof of Proposition 3.3

**Proposition 3.3.** *Let* $x \in \mathbb{R}$. *If* $\exp(-\mathcal{E}_t(x)) > 0$, $\exp(-\mathcal{E}(x_{0|t}^{(i)}))$ *is sub-Gaussian, and* $\|\nabla \exp(-\mathcal{E}(x_{0|t}^{(i)}))\|$ *is bounded. Then*

$$\frac{Var[S_K(x_t, t)]}{Var[E_K(x_t, t)]} = 4(1 + \|\nabla \mathcal{E}_t(x_t)\|)^2 + \mathcal{O}\left(\frac{1}{K}\right). \tag{39}$$

**Proof.**   Review that $S_K$ can be expressed as an importance-weighted estimator as follows:

$$S_K(x_t, t) = \frac{\frac{1}{K}\sum_{i=1}^{K}\nabla\exp(-\mathcal{E}(x_{0|t}^{(i)}))}{\frac{1}{K}\sum_{i=1}^{K}\exp(-\mathcal{E}(x_{0|t}^{(i)}))} \tag{40}$$

According to the assumpsion that $\|\nabla\exp(-\mathcal{E}(x_{0|t}^{(i)}))\|$ is bounded, let $\|\nabla\exp(-\mathcal{E}(x_{0|t}^{(i)}))\| \le M$, where $M > 0$. Since a bounded variable is sub-Gaussian, this assumption resembles a sub-Gaussianess assumption of $\|\nabla\exp(-\mathcal{E}(x_{0|t}^{(i)}))\|$.

On the other hand, according to the assumption that $\exp(-\mathcal{E}_t(x)) > 0$ for any $x$ in the support, there exists a constant $c$ such that $\exp(-\mathcal{E}(x_{0|t}^{(i)})) \ge c > 0$ and thus

$$\|S_K(x_t, t)\| = \left\|\frac{\frac{1}{K}\sum_{i=1}^{K}\nabla\exp(-\mathcal{E}(x_{0|t}^{(i)}))}{\frac{1}{K}\sum_{i=1}^{K}\exp(-\mathcal{E}(x_{0|t}^{(i)}))}\right\| \tag{41}$$

$$\le \frac{\|\sum_{i=1}^{K}\nabla\exp(-\mathcal{E}(x_{0|t}^{(i)}))\|}{Kc} \le M/c \tag{42}$$

Therefore, $S_K$ is bounded by $M/c$, suggesting it is sub-Gaussian. Recap that the error bound of $S_K$ is given by Eq. (34) as follows.

$$\|S_K(x_t, t) - S(x_t, t)\| \le \frac{2C\sqrt{\log(\frac{2}{\delta})}(1 + \|\nabla\mathcal{E}_t(x_t)\|)\exp(\mathcal{E}_t(x_t))}{\sqrt{K}} \tag{43}$$

which suggests that we can approximate the variance of $S_K(x_t, t)$, by leveraging its sub-Gaussianess, as follows,

$$\mathrm{Var}[S_K(x_t, t)] = \frac{4v_{0t}(x_t)(1 + \|\nabla\mathcal{E}_t(x_t)\|)^2}{m_t^2(x_t)K} + \mathcal{O}\left(\frac{1}{K^2}\right) \tag{44}$$

Therefore, according to Eq. (26) we can derive that

$$\frac{\mathrm{Var}[S_K(x_t, t)])}{\mathrm{Var}[E_K(x_t, t)]} = \frac{\frac{4v_{0t}(x_t)(1+\|\nabla\mathcal{E}_t(x_t)\|)^2}{m_t^2(x_t)K} + \mathcal{O}\left(\frac{1}{K^2}\right)}{\frac{v_{0t}(x_t)}{m_t^2(x_t)K} + \mathcal{O}\left(\frac{1}{K^2}\right)} \overset{(i)}{=} 4(1 + \|\nabla\mathcal{E}_t(x_t)\|)^2 + \mathcal{O}\left(\frac{1}{K}\right). \tag{45}$$

Step (i) follows from the asymptotic expansion of the ratio. By factoring out the leading order term $1/K$ from both the numerator and denominator, the quotient converges to the ratio of the coefficients with a residual error of order $\mathcal{O}(1/K)$. $\qquad\square$

## E   Bias and Error for Learned Scores

In our diffusion sampling setting, the most essential component is the learned score, which can be given by learning the MC score estimator (Eq. (7); iDEM) directly or differentiating a learned energy network (NEM and BNEM). In this section, we discuss the scores learned by iDEM and NEM, as well as their bias and error w.r.t. the ground truth noisy score $S_t(x_t) = -\nabla\mathcal{E}_t(x_t)$. In E.1 we first show that in optimal case, where the networks are infinitely capable and we train it for infinitely long, NEM would ultimately be equivalent to iDEM in terms of learned scores. Then we characterize the bias and error w.r.t. $S_t(x_t)$:

- Bias of Learned Scores (E.2): the optimal scores learned by NEM and iDEM, *i.e.* $\mathbb{E}[E_K(x_t, t)]$, are biased due to the bias of the learning targets $E_K$ (for NEM) and $S_K$ (for iDEM). It results in inreducible errors to $S_t(x_t)$.

- Error of Learned Scores (E.3): the neural network is theoretically guaranteed to converge to the optimal value, *i.e.* $\mathbb{E}[E_K(x_t, t)]$ for NEM and $\mathbb{E}[S_K(x_t, t)]$ for iDEM, with infinitely flexible network and infinite training time. While in practice, the learned networks are often imprefect. It results in non-zero learning errors in practice, which are theoretically reducible.

### E.1 iDEM≡NEM in optimal case

Suppose both the score network $s_\phi(x_t, t)$ and the energy network $E_\theta(x_t, t)$ are capable enough and we train both networks infinitely long. Then the optimal value they learn at $(x_t, t)$ would be the expected value of their training targets, *i.e.* $s_{\phi^*}(x_t, t) = \mathbb{E}[S_K(x_t, t)]$ and $E_{\theta^*}(x_t, t) = \mathbb{E}[E_K(x_t, t)]$. Notice that $S_K(x_t, t) = -\nabla_{x_t} E_K(x_t, t)$, we can rewrite the optimal learned scores given by the energy network as follows:

$$-\nabla_{x_t} E_{\theta^*}(x_t, t) = -\nabla_{x_t} \mathbb{E}[E_K(x_t, t)] \tag{46}$$

$$= -\nabla_{x_t} \mathbb{E}_{x_{0|t}^{(1:K)}} \left[ \log \frac{1}{K} \sum_{i=1}^{K} \exp\left(-\mathcal{E}(x_{0|t}^{(i)})\right) \right], \quad x_{0|t}^{(i)} \sim \mathcal{N}(x; x_t, \sigma_t^2 I) \tag{47}$$

$$= -\nabla_{x_t} \mathbb{E}_{\epsilon_{0|t}^{(1:K)}} \left[ \log \frac{1}{K} \sum_{i=1}^{K} \exp\left(-\mathcal{E}(x_t + \epsilon_{0|t}^{(i)})\right) \right], \quad \epsilon_{0|t}^{(i)} \sim \mathcal{N}(\epsilon; 0, \sigma_t^2 I) \tag{48}$$

$$= \mathbb{E}_{\epsilon_{0|t}^{(1:K)}} \left[ -\nabla_{x_t} \log \frac{1}{K} \sum_{i=1}^{K} \exp\left(-\mathcal{E}(x_t + \epsilon_{0|t}^{(i)})\right) \right] \tag{49}$$

$$= \mathbb{E}_{x_{0|t}^{(1:K)}} \left[ -\nabla_{x_t} \log \frac{1}{K} \sum_{i=1}^{K} \exp\left(-\mathcal{E}\left(x_{0|t}^{(i)}\right)\right) \right] \tag{50}$$

$$= \mathbb{E}[-\nabla_{x_t} E_K(x_t, t)] \tag{51}$$

$$= s_{\phi^*}(x_t, t) \tag{52}$$

Therefore, in the optimal case, scores learned by both NEM and iDEM would be equal. Then a question arises as to "*Why is NEM more favorable theoretically and practically, without concern about the additional computation?*". We will address this question after presenting the bias for the learned scores. Notice that the above derivation also shows that, to compute the expected value of $S_K$, one can first compute $\mathbb{E}[E_K(x_t, t)]$ then differentiate this expectation w.r.t. $x_t$.

### E.2 Bias of Learned Scores

Since the regressing targets are biased, the optimal learned scores are biased. To derive the bias term, we assume the sub-Gaussianess of $\mathcal{E}(x_{0|t}^{(i)})$ and leverage Eq. (46). According to Proposition 3.1 and Corollary 3.2, the optimal energy network is obtained as follows

$$E_{\theta^*}(x_t, t) = \mathcal{E}_t(x_t) + \frac{v_{0t}(x_t)}{2 m_t^2(x_t) K} \tag{53}$$

We first review $m_t(x_t)$ (Eq. (22)) and $v_{0t}(x_t)$ (Eq. (23)):

$$m_t(x_t) = \exp(-\mathcal{E}_t(x_t)) \tag{54}$$

$$v_{0t}(x_t) = \text{Var}_{\mathcal{N}(x; x_t, \sigma_t^2 I)}[\exp(-\mathcal{E}(x))] \tag{55}$$

$$= \int \exp(-2\mathcal{E}(x)) \mathcal{N}(x_t; x, \sigma_t^2 I) dx - m_t^2(x_t) \tag{56}$$

We can derive their derivatives w.r.t. $x_t$ as follows:

$$m_t(x_t)' = -m_t(x_t)\nabla\mathcal{E}_t(x_t) \tag{57}$$

$$v_{0t}(x_t)' = \nabla_{x_t}\int \exp(-2\mathcal{E}(x))\mathcal{N}(x_t; x, \sigma_t^2 I)dx - 2m_t(x_t)m_t'(x_t) \tag{58}$$

$$= \int \exp(-2\mathcal{E}(x))\nabla_{x_t}\mathcal{N}(x_t; x, \sigma_t^2 I)dx - 2m_t(x_t)m_t'(x_t) \tag{59}$$

$$= \int -\exp(-2\mathcal{E}(x))\nabla_x\mathcal{N}(x_t; x, \sigma_t^2 I)dx - 2m_t(x_t)m_t'(x_t) \tag{60}$$

$$= -\exp(-2\mathcal{E}(x))\mathcal{N}(x_t; x, \sigma_t^2 I)|_{x=-\infty}^{x=+\infty} +$$
$$\int \nabla_x \exp(-2\mathcal{E}(x))\mathcal{N}(x_t; x, \sigma_t^2 I)dx - 2m_t(x_t)m_t'(x_t) \tag{61}$$

$$= \int \nabla_x \exp(-2\mathcal{E}(x))\mathcal{N}(x_t; x, \sigma_t^2 I)dx - 2m_t(x_t)m_t'(x_t) \tag{62}$$

$$= -2\int \exp(-2\mathcal{E}(x))\nabla_x\mathcal{E}(x)\mathcal{N}(x_t; x, \sigma_t^2 I)dx + 2m_t^2(x_t)\nabla\mathcal{E}_t(x_t) \tag{63}$$

Therefore, the predicted noisy score at $(x_t, t)$ is given by differentiating Eq. (53) w.r.t. $x_t$, which is

$$-\nabla_{x_t}\left(\mathcal{E}_t(x_t) + \frac{v_{0t}(x_t)}{2m_t^2(x_t)K}\right) \tag{64}$$

$$= S_t(x_t) - \frac{1}{2K}\frac{v_{0t}'(x_t)m_t^2(x_t) - 2v_{0t}(x_t)m_t(x_t)m_t'(x_t)}{m_t^4(x_t)} \tag{65}$$

$$= S_t(x_t) - \frac{v_{0t}'(x_t) + 2v_{0t}(x_t)\nabla\mathcal{E}_t(x_t)}{2m_t^2(x_t)K} \tag{66}$$

$$= S_t(x_t) - \frac{1}{m_t^2(x_t)K}\left(\int \exp(-2\mathcal{E}(x))\left(\nabla_{x_t}\mathcal{E}_t(x_t) - \nabla_x\mathcal{E}(x)\right)\mathcal{N}(x_t; x, \sigma_t^2 I)dx + m_t^2(x_t)\nabla\mathcal{E}_t(x_t)\right) \tag{67}$$

$$= \left(1 + \frac{1}{K}\right)S_t(x_t) - \frac{1}{K}\int \exp(-2\mathcal{E}(x) + 2\mathcal{E}_t(x_t))\left(\nabla_{x_t}\mathcal{E}_t(x_t) - \nabla_x\mathcal{E}(x)\right)\mathcal{N}(x_t; x, \sigma_t^2 I)dx \tag{68}$$

$$= \left(1 + \frac{1}{K}\right)S_t(x_t) + \frac{1}{2K}\int \nabla_{x_t}\exp(2\mathcal{E}_t(x_t) - 2\mathcal{E}(x_t + \epsilon))\mathcal{N}(\epsilon; 0, I)d\epsilon \tag{69}$$

Therefore, we can derive the bias of learned score as follows:

$$\|-\nabla_{x_t}E_\theta^*(x_t, t) - S_t(x_t)\| \tag{70}$$

$$= \left\|\frac{1}{K}S_t(x_t) - \frac{1}{K}\int \exp(-2\mathcal{E}(x) + 2\mathcal{E}_t(x_t))\left(\nabla_{x_t}\mathcal{E}_t(x_t) - \nabla_x\mathcal{E}(x)\right)\mathcal{N}(x_t; x, \sigma_t^2 I)dx\right\| \tag{71}$$

Eq. (71) shows that the score bias decreases linearly w.r.t. number of MC samples $K$. In particular, at low noise level (*i.e.* small $t$), the latter term of Eq. (68) or Eq. (69) would vanish as (1) $\mathcal{E}_t(x)$ is close to $\mathcal{E}(x)$ and (2) $x \sim \mathcal{N}(x_t, \sigma_t^2 I)$ are close to $x_t$, resulting in a small bias of the scores given by both differentiating the learned energy network and the directly learned score network.

For BNEM, given a bootstrapping schedule $\{s_i\}_{i=1}^n$ with $s_0 = 0$ and $s_n = 1$, the optimal network is given by

$$E_{\theta^{**}}(x_t, t) = \mathcal{E}_t(x_t) + \frac{v_{0t}(x_t)}{2m_t^2(x_t)K^{i+1}} + \sum_{j=1}^i \frac{v_{0s_j}(x_t)}{2m_{s_j}^2(x_t)K^j} \tag{72}$$

where $t \in [s_i, s_{i+1}]$. Then the learned score is obtained by differentiating $E_{\theta^{**}}$

$$-\nabla_{x_t} E_{\theta^{**}}(x_t, t) = -\nabla_{x_t} \left( \mathcal{E}_t(x_t) + \frac{v_{0t}(x_t)}{2m_t^2(x_t)K^{i+1}} + \sum_{j=1}^{i} \frac{v_{0s_j}(x_t)}{2m_{s_j}^2(x_t)K^j} \right) \tag{73}$$

$$= \left( 1 + \sum_{j=1}^{i+1} \frac{1}{K^j} \right) S_t(x_t) \tag{74}$$

$$- \frac{1}{K^{i+1}} \int \exp(-2\mathcal{E}(x) + 2\mathcal{E}_t(x_t)) \left( \nabla_{x_t} \mathcal{E}_t(x_t) - \nabla_x \mathcal{E}(x) \right) \mathcal{N}(x_t; x, \sigma_t^2 I) dx$$

$$- \sum_{j=1}^{i} \frac{1}{K^j} \int \exp(-2\mathcal{E}(x) + 2\mathcal{E}_{s_j}(x_t)) \left( \nabla_{x_t} \mathcal{E}_{s_j}(x_t) - \nabla_x \mathcal{E}(x) \right) \mathcal{N}(x_t; x, \sigma_{s_j}^2 I) dx$$

which gives us the bias of learned scores for BNEM.

### E.3 Error of Learned Scores

Even though the neural networks can converge to the optimal value in theory by assuming infinite training time and infinitely flexible architecture, they are always imperfect in practice. To measure this imperfectness, we define the "learning error" as the discrepancy between the actual learned value and the optimal learned value. To address the question "*Why is NEM more favorable theoretically and practically, without concern about the additional computation?*", we empirically claim that regressing the energy estimator $E_K$ is an easier learning task against regressing the score estimator $S_K$ as the former target is shown to have smaller variance in Proposition 3.3, resulting in smaller learning error.

Let $E_\theta(x_t, t) = \mathbb{E}[E_K(x_t, t)] + e_{\mathcal{E}}(x_t, t)$ be the learned energy network and $S_\theta(x_t, t) = \mathbb{E}[S_K(x_t, t)] + e_S(x_t, t)$ be the learned score network, where $e_{\mathcal{E}}$ and $e_S$ are the corresponding learning errors. We also assume that $e_{\mathcal{E}}$ is differentiable w.r.t. $x_t$. Then the score given by the energy network is

$$\nabla_{x_t} E_\theta(x_t, t) = \nabla_{x_t} \mathbb{E}[E_K(x_t, t)] + \nabla_{x_t} e_{\mathcal{E}}(x_t, t) \tag{75}$$

$$= \mathbb{E}[S_K(x_t, t)] + \nabla_{x_t} e_{\mathcal{E}}(x_t, t) \tag{76}$$

Therefore, the errors of the network that affects the diffusion sampling are $\nabla_{x_t} e_{\mathcal{E}}(x_t, t)$ (for NEM) and $e_S(x_t, t)$ (for iDEM). Since learning the energy is easier due to the small variance of its training target, we can assume that $e_{\mathcal{E}}(x_t, t)$ is $L$-Lipchitz. Therefore, the differentiated learning error of the energy network is controlled by the Lipchitz constant $L$, *i.e.*, $\|\nabla_{x_t} e_{\mathcal{E}}(x_t, t)\| \leq L$.

In practice, this Lipchitz constant can be small because learning the energy estimator is an easier task compared to learning the score estimator, resulting in $L < e_S(x_t, t)$, which supports the empirical experiments that NEM performs better than iDEM.

## F Ideal Bootstrap Estimator $\equiv$ Recursive Estimator

In this section, we will show that an ideal bootstrap estimator can polynomially reduce both the variance and bias w.r.t. number of MC samples $K$. Let a local bootstrapping trajectory $u \to s \to t$, with $u < s < t$. Suppose we have access to $\mathcal{E}_u$. Let $E_{\tilde{\theta}}(x_s, s)$ is learned from targetting an estimator bootstrapped from $\mathcal{E}_u$ to approximate $\mathcal{E}_s$. Suppose $E_{\tilde{\theta}}(x_s, s)$ is ideal, *i.e.* $E_{\tilde{\theta}}(x_s, s) \equiv E_K(x_s, s, u; \mathcal{E}_u)$. The bootstrap energy estimator (Eq. (15)) at $t$ from $s$ can be written as follows:

$$E_K(x_t, t, s; \tilde{\theta}) = -\log \frac{1}{K} \sum_{i=1}^{K} \exp(-E_{\tilde{\theta}}(x_{s|t}^{(i)}, s)), \quad x_{s|t}^{(i)} \sim \mathcal{N}(x; x_t, (\sigma_t^2 - \sigma_s^2)I) \tag{77}$$

By plugging $E_{\tilde{\theta}}(x_{s|t}^{(i)}, s) = E_K(x_{s|t}^{(i)}, s, u; \mathcal{E}_u)$ into the above equation, we have

$$E_K(x_t, t, s; \tilde{\theta}) = -\log \frac{1}{K} \sum_{i=1}^{K} \exp(-E_K(x_{s|t}^{(i)}, s, u; \mathcal{E}_u)) \tag{78}$$

$$= -\log \frac{1}{K} \sum_{i=1}^{K} \frac{1}{K} \sum_{j=1}^{K} \exp(-\mathcal{E}_u(x_{u|s}^{(ij)})), \quad x_{u|s}^{(ij)} \sim \mathcal{N}(x; x_s^{(i)}, (\sigma_s^2 - \sigma_u^2)I) \tag{79}$$

where $x_{u|s}^{(ij)} \sim \mathcal{N}(x; x_s^{(i)}, (\sigma_s^2 - \sigma_u^2)I)$. Notice that $x_{s|t}^{(i)} = x_t + \sqrt{\sigma_t^2 - \sigma_s^2}\epsilon_s^{(i)}$, $x_{u|s}^{(ij)} = x_s^{(i)} + \sqrt{\sigma_s^2 - \sigma_u^2}\epsilon_u^{(ij)}$, and $\epsilon_s^{(j)}$ and $\epsilon_u^{(ij)}$ are independent, we can combine these Gaussian-noise injection:

$$E_K(x_t, t, s; \tilde{\theta}) = -\log \frac{1}{K^2} \sum_{i=1}^{K} \sum_{j=1}^{K} \exp(-\mathcal{E}_u(x_{u|t}^{(ij)})) = E_{K^2}(x_t, t, u; \mathcal{E}_u) \tag{80}$$

where $x_{u|t}^{(ij)} \sim \mathcal{N}(x; x_t, (\sigma_t^2 - \sigma_u^2)I)$. Therefore, the ideal bootstrapping at $t$ from $s$ over this local trajectory $(u \to s \to t)$ is equivalent to using quadratically more samples compared with estimating $t$ directly from $u$. With initial condition $\mathcal{E}_0 = \mathcal{E}$ and $\sigma_0 = 0$, we can simply show that: Given a bootstrap trajectory $\{s_i\}_{i=0}^{n}$ where $s_0 = 0$, $s_n = 1$. For any $t \in [s_i, s_{i+1}]$ and any $s \in [s_{i-1}, s_i]$, with $i = 0, ..., n - 1$, the **ideal** bootstrap estimator is equivalent to

$$E_K(x_t, t, s; \tilde{\theta}) \equiv E_{K^{(i+1)}}(x_t, t) \tag{81}$$

which polynomially reduces the variance and bias of the estimator. In other words, the ideal bootstrap estimator is equivalent to recursively approximating $\mathcal{E}_s$ with $E_K(x_s, s)$ utill the initial condition $\mathcal{E}$.

For simplification in appendix G, we term the ideal bootstrap estimator with bootstrapping $n$ times as the **Sequential**$(n)$ **Estimator**, $E^{\text{Seq}(n)}$, *i.e.*

$$E_K^{\text{Seq}(n)}(x_t, t) := E_{K^{n+1}}(x_t, t) \tag{82}$$

# G  Proof of Proposition 3.4

**Proposition 3.4.** *Given a bootstrap trajectory $\{s_i\}_{i=0}^{n}$ such that $\sigma_{s_i}^2 - \sigma_{s_{i-1}}^2 \leq \kappa$, where $s_0 = 0$, $s_n = 1$ and $\kappa > 0$ is a small constant. Suppose $E_\theta$ is incrementally optimized from $t \in [0, 1]$ as follows: if $t \in [s_i, s_{i+1}]$, $E_\theta(x_t, t)$ targets an energy estimator bootstrapped from $\forall s \in [s_{i-1}, s_i]$ using Eq. (15). For $\forall 0 \leq i \leq n$ and $\forall s \in [s_{i-1}, s_i]$, the variance of the bootstrap energy estimator is approximately given by*

$$Var[E_K(x_t, t, s; \theta)] = \frac{v_{st}(x_t)}{v_{0t}(x_t)} Var[E_K(x_t, t)] \tag{83}$$

*and the bias of $E_K(x_t, t, s; \theta)$ is approximately given by*

$$\text{Bias}[E_K(x_t, t, s; \theta)] = \frac{v_{0t}(x_t)}{2m_t^2(x_t)K^{i+1}} + \sum_{j=1}^{i} \frac{v_{0s_j}(x_t)}{2m_{s_j}^2(x_t)K^j}. \tag{84}$$

*where $m_z(x_z) = \exp(-\mathcal{E}_z(x_z))$ and $v_{yz}(x_z) = \text{Var}_{\mathcal{N}(x; x_z, (\sigma_z^2 - \sigma_y^2)I)}[\exp(-\mathcal{E}_y(x))]$ for $\forall 0 \leq y < z \leq 1$.*

***Proof.*** The variance of $E_K(x_t, t, s; \theta)$ can be simply derived by leveraging the variance of a sub-Gaussian random variable similar to Eq. (26). While the entire proof for bias of $E_K(x_t, t, s; \theta)$ is organized as follows:

1. we first show the bias of Bootstrap(1) estimator, which is bootstrapped from the system energy

2. we then show the bias of Bootstrap($n$) estimator, which is bootstrapped from a lower level noise convolved energy recursively, by induction.

### G.1 Bootstrap(1) estimator

The Sequential estimator and Bootstrap(1) estimator are defined by:

$$E_K^{\text{Seq}(1)}(x_t, t) := -\log \frac{1}{K} \sum_{i=1}^{K} \exp(-E_K(x_{s|t}^{(i)}, s)), \quad x_{s|t}^{(i)} \sim \mathcal{N}(x; x_t, (\sigma_t^2 - \sigma_s^2)I) \tag{85}$$

$$= -\log \frac{1}{K^2} \sum_{i=1}^{K} \sum_{j=1}^{K} \exp(-\mathcal{E}(x_{0|t}^{(ij)})), \quad x_{0|t}^{(ij)} \sim \mathcal{N}(x; x_t, \sigma_t^2 I) \tag{86}$$

$$E_K^{B(1)}(x_t, t, s; \theta) := -\log \frac{1}{K} \sum_{i=1}^{K} \exp(-E_\theta(x_{s|t}^{(i)}, s)), \quad x_{s|t}^{(i)} \sim \mathcal{N}(x; x_t, (\sigma_t^2 - \sigma_s^2)I) \tag{87}$$

The mean and variance of a Sequential estimator can be derived by considering it as the MC estimator with $K^2$ samples:

$$\mathbb{E}[E_K^{\text{Seq}(1)}(x_t, t)] = \mathcal{E}_t(x_t) + \frac{v_{0t}(x_t)}{2m_{0t}^2(x_t)K^2} \quad \text{and} \quad \text{Var}(E_K^{\text{Seq}(1)}(x_t, t)) = \frac{v_{0t}(x_t)}{m_{0t}^2(x_t)K^2} \tag{88}$$

While an optimal network obtained by targeting the original MC energy estimator Eq. (9) at $s$ is [4] :

$$E_{\theta^*}(x_s, s) = \mathbb{E}[E_K(x_s, s)] = -\log m_s(x_s) + \frac{v_{0s}(x_s)}{2m_s^2(x_s)K} \tag{89}$$

Then the optimal Bootstrap(1) estimator can be expressed as:

$$E_K^{B(1)}(x_t, t, s; \theta^*) = -\log \frac{1}{K} \sum_{i=1}^{K} \exp\left(-\left(-\log m_s(x_{s|t}^{(i)}) + \frac{v_{0s}(x_{s|t}^{(i)})}{2m_s^2(x_{s|t}^{(i)})K}\right)\right) \tag{90}$$

Before linking the Bootstrap estimator and the Sequential one, we provide the following approximation which is useful. Let $a$, $b$ two random variables and $\{a_i\}_{i=1}^K$, $\{b_i\}_{i=1}^K$ are corresponding samples. Assume that $\{b_i\}_{i=1}^K$ are close to 0 and concentrated at $m_b$ (hence $m_b$ closes to 0), while $\{a_i\}_{i=1}^K$ are concentrated at $m_a$, then

$$\log \frac{1}{K} \sum_{i=1}^{K} \exp(-(a_i + b_i)) = \log \frac{1}{K} \left\{ \sum_{i=1}^{K} \exp(-a_i) \left[ \frac{\sum_{i=1}^{K} \exp(-(a_i + b_i))}{\sum_{i=1}^{K} \exp(-a_i)} \right] \right\} \tag{91}$$

$$= \log \frac{1}{K} \sum_{i=1}^{K} \exp(-a_i) + \log \frac{\sum_{i=1}^{K} \exp(-(a_i + b_i))}{\sum_{i=1}^{K} \exp(-a_i)} \tag{92}$$

$$= \log \frac{1}{K} \sum_{i=1}^{K} \exp(-a_i) + \log \frac{\sum_{i=1}^{K} \exp(-a_i)(1 - b_i)}{\sum_{i=1}^{K} \exp(-a_i)} + \mathcal{O}(\max_i |b_i|^2) \tag{93}$$

$$= \log \frac{1}{K} \sum_{i=1}^{K} \exp(-a_i) + \log \left(1 - \frac{\sum_{i=1}^{K} \exp(-a_i)b_i}{\sum_{i=1}^{K} \exp(-a_i)}\right) + \mathcal{O}(\max_i |b_i|^2) \tag{94}$$

$$= \log \frac{1}{K} \sum_{i=1}^{K} \exp(-a_i) - \frac{\sum_{i=1}^{K} \exp(-a_i)b_i}{\sum_{i=1}^{K} \exp(-a_i)} + \mathcal{O}(\max_i |b_i|^2) \tag{95}$$

$$= \log \frac{1}{K} \sum_{i=1}^{K} \exp(-a_i) - m_b + \mathcal{O}(\max_i |b_i|^2) \tag{96}$$

where approximation Eq. (93) applies a first order Taylor expansion of $e^x \approx 1 + x$ around $x = 0$ since $\{b_i\}_{i=1}^K$ are close to 0; while Approximation uses $\log(1 + x) \approx x$ under the same assumption. Notice that when $K$

---

[4]We consider minimizing the $L_2$-norm, *i.e.* $\theta^* = \arg\min_\theta \mathbb{E}_{x_0, t}[\|E_\theta(x_t, t) - E_K(x_t, t)\|^2]$. Since the target, $E_K$, is noisy, the optimal outputs are given by the expectation, *i.e.* $E_\theta^* = \mathbb{E}[E_K]$.

is large and $\Delta\sigma_{st}^2 := \sigma_t^2 - \sigma_s^2 \leq \kappa$ is small , $\{\frac{v_{0s}(x_{s|t}^{(i)})}{2m_s^2(x_{s|t}^{(i)})K}\}_{i=1}^K$ are close to 0 and concentrated at $\frac{v_{0s}(x_t)}{2m_s^2(x_t)K}$ . Therefore, by plugging them into Eq. (96), Eq. (90) can be approximated by

$$E_K^{B(1)}(x_t, t, s; \theta^*) \approx -\log \frac{1}{K} \sum_{i=1}^K m_s(x_{s|t}^{(i)}) + \frac{v_{0s}(x_t)}{2m_s^2(x_t)K}. \tag{97}$$

When $K$ is large and $\sigma_t^2 - \sigma_s^2$ is small, the bias and variance of $E_K(x_{s|t}^{(i)}, s)$ are small, then we have

$$-\log \frac{1}{K} \sum_{i=1}^K m_s(x_{s|t}^{(i)}) \approx -\log \frac{1}{K} \sum_{i=1}^K \exp(-E_K(x_{s|t}^{(i)}, s)) = E_K^{\text{Seq}(1)}(x_t, t). \tag{98}$$

Therefore, the optimal Bootstrap estimator can be approximated as follows:

$$E_K^{B(1)}(x_t, t, s; \theta^*) \approx E_K^{\text{Seq}(1)}(x_t, t) + \frac{v_{0s}(x_t)}{2m_s^2(x_t)K}, \tag{99}$$

where its mean and variance depend on those of the Sequential estimator (Eq. (88)):

$$\mathbb{E}[E_K^{B(1)}(x_t, t, s; \theta^*)] = \mathcal{E}_t(x_t) + \frac{v_{0t}(x_t)}{2m_t^2(x_t)K^2} + \frac{v_{0s}(x_t)}{2m_s^2(x_t)K}, \tag{100}$$

$$\text{Var}[E_K^{B(1)}(x_t, t, s; \theta^*)] = \frac{v_{0t}(x_t)}{m_t^2(x_t)K^2}. \tag{101}$$

## G.2 Bootstrap($n$) estimator

Given a bootstrap trajectory $\{s_i\}_{i=1}^n$ where $s_0 = 0$ and $s_n = s$, and $E_\theta$ is well learned at $[0, s]$. Let the energy network be optimal for $u \leq s_n$ by learning a sequence of Bootstrap($i$) energy estimators ($i \leq n$). Then the optimal value of $E_\theta(x_s, s)$ is given by $\mathbb{E}[E_K^{B(n-1)}(x_s, s)]$. We are going to show the variance of a Bootstrap($n$) estimator by induction. Suppose we have:

$$E_{\theta^*}(x_s, s) = \mathcal{E}_s(x_s) + \sum_{j=1}^n \frac{v_{0s_j}(x_s)}{2m_{s_j}^2(x_s)K^j} \tag{102}$$

$$= \mathbb{E}[E_K^{\text{Seq}(n-1)}(x_s, s)] + \sum_{j=1}^{n-1} \frac{v_{0s_j}(x_s)}{2m_{s_j}^2(x_s)K^j} \tag{103}$$

Then for any $t \in (s, 1]$, the learning target of $E_\theta(x_t, t)$ is bootstrapped from $s_n = s$,

$$E_K^{B(n)}(x_t, t) = -\log \frac{1}{K} \sum_{i=1}^K \exp(-E_{\theta^*}(x_{s|t}^{(i)}, s)), \quad x_{s|t}^{(i)} \sim \mathcal{N}(x; x_t, (\sigma_t^2 - \sigma_s^2)I) \tag{104}$$

$$= -\log \frac{1}{K} \sum_{i=1}^K \exp\left(-\mathbb{E}[E_K^{\text{Seq}(n-1)}(x_{s|t}^{(i)}, s)] - \sum_{j=1}^{n-1} \frac{v_{0s_j}(x_{s|t}^{(i)})}{2m_{s_j}^2(x_{s|t}^{(i)})K^j}\right) \tag{105}$$

Assume that $\sigma_t^2 - \sigma_s^2$ is small and $K$ is large, then we can apply Approximation Eq. (96) and have

$$E_K^{B(n)}(x_t, t) = -\log \frac{1}{K} \sum_{i=1}^K \exp\left(-\mathbb{E}[E_K^{\text{Seq}(n-1)}(x_{s|t}^{(i)}, s)]\right) + \sum_{j=1}^{n-1} \frac{v_{0s_j}(x_t)}{2m_{s_j}^2(x_t)K^j} \tag{106}$$

The first term in the RHS of the above equation can be further simplified as [5] :

$$-\log\frac{1}{K}\sum_{i=1}^{K}\exp\left(-\mathbb{E}[E_K^{\mathrm{Seq}(n-1)}(x_{s|t}^{(i)},s)]\right) \tag{107}$$

$$=\mathbb{E}\left[-\log\frac{1}{K}\sum_{i=1}^{K}\exp\left(-E_K^{\mathrm{Seq}(n-1)}(x_{s|t}^{(i)},s)\right)\right]+\frac{\mathrm{Var}[E_K^{\mathrm{Seq}(n-1)}(x_{s|t}^{(i)},s)]}{2}+\mathcal{O}\left(\frac{\mathrm{Var}[E_K^{\mathrm{Seq}(n-1)}(x_{s|t}^{(i)},s)]}{K}\right) \tag{108}$$

$$=\mathbb{E}_{\epsilon^{(j)}}\left[-\log\frac{1}{K^{n+1}}\sum_{i=1}^{K}\sum_{j=1}^{K^n}\exp\left(-\mathcal{E}\left(x_{s|t}^{(i)}+\epsilon^{(j)}\right)\right)\right]+\frac{v_{0s}(x_t)}{2m_s^2(x_t)K^n} \tag{109}$$

where $\epsilon^j \overset{\text{i.i.d.}}{\sim} \mathcal{N}(0,I)$. Therefore, Eq. (106) can be simplified as

$$E_K^{B(n)}(x_t,t) = \mathbb{E}_{\epsilon^{(j)}}\left[-\log\frac{1}{K^{n+1}}\sum_{i=1}^{K}\sum_{j=1}^{K^n}\exp\left(-\mathcal{E}\left(x_{s|t}^{(i)}+\epsilon^{(j)}\right)\right)\right]+\sum_{j=1}^{n}\frac{v_{0s_j}(x_t)}{2m_{s_j}^2(x_t)K^j} \tag{110}$$

Taking the expectation over Eq. (110) yields an expectation of the Sequential($n$) estimator on its RHS, we therefore complete the proof by induction:

$$\mathbb{E}\left[E_K^{B(n)}(x_t,t)\right] = \mathbb{E}_{\epsilon^{(j)},x_{s|t}^{(i)}}\left[-\log\frac{1}{K^{n+1}}\sum_{i=1}^{K}\sum_{j=1}^{K^n}\exp\left(-\mathcal{E}\left(x_{s|t}^{(i)}+\epsilon^{(j)}\right)\right)\right]+\sum_{j=1}^{n}\frac{v_{0s_j}(x_t)}{2m_{s_j}^2(x_t)K^j} \tag{111}$$

$$=\mathbb{E}\left[E_K^{\mathrm{Seq}(n)}(x_t,t)\right]+\sum_{j=1}^{n}\frac{v_{0s_j}(x_t)}{2m_{s_j}^2(x_t)K^j} \tag{112}$$

$$=\mathcal{E}_t(x_t)+\frac{v_{0t}(x_t)}{2m_t^2(x_t)K^{n+1}}+\sum_{j=1}^{n}\frac{v_{0s_j}(x_t)}{2m_{s_j}^2(x_t)K^j} \tag{113}$$

which suggests that the accumulated bias of a Bootstrap($n$) estimator is given by

$$\mathrm{Bias}[E_K^{B(n)}(x_t,t)] = \frac{v_{0t}(x_t)}{2m_t^2(x_t)K^{n+1}}+\sum_{j=1}^{n}\frac{v_{0s_j}(x_t)}{2m_{s_j}^2(x_t)K^j} \tag{114}$$

$$\square$$

## H   Incorporating Symmetry Using NEM

We consider applying NEM and BNEM in physical systems with symmetry constraints like $n$-body system. We prove that our MC energy estimator $E_K$ is $G$-invariant under certain conditions, given in the following Proposition.

**Proposition H.1.** *Let $G$ be the product group $SE(3) \times \mathbb{S}_n \hookrightarrow O(3n)$ and $p_0$ be a $G$-invariant density in $\mathbb{R}^d$. Then the Monte Carlo energy estimator of $E_K(x_t,t)$ is $G$-invariant if the sampling distribution $x_{0|t} \sim \mathcal{N}(x_{0|t}; x_t, \sigma_t^2)$ is $G$-invariant, i.e.,*

$$\mathcal{N}(x_{0|t}; g \circ x_t, \sigma_t^2) = \mathcal{N}(g^{-1}x_{0|t}; x_t, \sigma_t^2).$$

---

[5]This simplication leverages the fact that, if $X_i$ are i.i.d. bounded sub-Gaussian random variables, then $\log\sum_{i=1}^{K}\exp(-\mathbb{E}X_i)$ can be approximated by $\mathbb{E}[\log\sum_{i=1}^{K}\exp(-X_i)] - \frac{1}{2}\mathrm{Var}(X) + \mathcal{O}\left(\mathrm{Var}(X)/K\right)$ .

**Proof.** Since $p_0$ is $G$-invariant, then $\mathcal{E}$ is $G$-invariant as well. Let $g \in G$ acts on $x \in \mathbb{R}^d$ where $g \circ x = gx$. Since $x_{0|t}^{(i)} \sim \overline{\mathcal{N}}(x_{0|t}; x_t, \sigma_t^2)$ is equivalent to $g \circ x_{0|t}^{(i)} \sim \overline{\mathcal{N}}(x_{0|t}; g \circ x_t, \sigma_t^2)$. Then we have

$$E_K(g \circ x_t, t) = -\log \frac{1}{K} \sum_{i=1}^{K} \exp(-\mathcal{E}(g \circ x_{0|t}^{(i)})) \tag{115}$$

$$= -\log \frac{1}{K} \sum_{i=1}^{K} \exp(-\mathcal{E}(x_{0|t}^{(i)})) = E_K(x_t, t) \tag{116}$$

$$x_{(0|t)}^{(i)} \sim \overline{\mathcal{N}}(x_{0|t}; x_t, \sigma_t^2) \tag{117}$$

Therefore, $E_K$ is invariant to $G = \mathrm{SE}(3) \times \mathbb{S}_n$. $\qquad \square$

Furthermore, $E_K(x_t, t, s; \phi)$ is obtained by applying a learned energy network, which is $G$-invariant, to the analogous process and therefore is $G$-invariant as well.

# I   Generalizing NEM

This section generalizes NEM in two ways: we first generalize the SDE setting, by considering a broader family of SDEs applied to sampling from Boltzmann distribution; then we generalize the MC energy estimator by viewing it as an importance-weighted estimator.

## I.1   NEM for General SDEs

Diffusion models can be generalized to any SDEs as $dx_t = f(x_t, t)dt + g(t)dw_t$, where $t \geq 0$ and $w_t$ is a Brownian motion. Particularly, we consider $f(x, t) := -\alpha(t)x$, *i.e.*

$$dx_t = -\alpha(t)x_t dt + g(t)dw_t \tag{118}$$

Then the marginal of the above SDE can be analytically derived as:

$$x_t = \beta(t)x_0 + \beta(t)\sqrt{\int_0^t (g(s)\beta(s))^2 ds}\,\epsilon, \quad \beta(t) := e^{-\int_0^t \alpha(s)ds} \tag{119}$$

where $\epsilon \sim \mathcal{N}(0, I)$. For example, when $g(t) = \sqrt{\bar{\beta}(t)}$ and $\alpha(t) = \frac{1}{2}\bar{\beta}(t)$, where $\bar{\beta}(t)$ is a monotonic function (*e.g.* linear) increasing from $\bar{\beta}_{\min}$ to $\bar{\beta}_{\max}$, the above SDE resembles a Variance Preserving (VP) process (Song et al., 2020). In DMs, VP can be a favor since it constrains the magnitude of noisy data across $t$; while a VE process doesn't, and the magnitude of data can explode as the noise explodes. Therefore, we aim to discover whether any SDEs rather than VE can be better by generalizing NEM and DEM to general SDEs.

In this work, we provide a solution for general SDEs (Eq. (118)) rather than a VE SDE. For simplification, we exchangeably use $\beta(t)$ and $\beta_t$. Given a SDE as Eq. (118) for any integrable functions $\alpha$ and $g$, we can first derive its marginal as Eq. (119), which can be expressed as:

$$\beta_t^{-1}x_t = x_0 + \sqrt{\int_0^t (g(s)\beta(s))^2 ds}\,\epsilon \tag{120}$$

Therefore, by defining $y_t = \beta_t^{-1}x_t$ we have $y_0 = x_0$ and therefore:

$$y_t = y_0 + \sqrt{\int_0^t (g(s)\beta(s))^2 ds}\,\epsilon \tag{121}$$

which resembles a VE SDE with noise schedule $\tilde{\sigma}^2(t) = \int_0^t (g(s)\beta(s))^2 ds$. We can also derive this by changing variables:

$$dy_t = (\beta^{-1}(t))' x_t dt + \beta^{-1}(t) dx_t \tag{122}$$

$$= \beta^{-1}(t)\alpha(t) x_t dt + \beta^{-1}(t)(-\alpha(t)x_t dt + g(t)dw_t) \tag{123}$$

$$= \beta^{-1}(t)g(t)dw_t \tag{124}$$

which also leads to Eq. (121). Let $\tilde{p}_t$ be the marginal distribution of $y_t$ and $p_t$ the marginal distribution of $x_t$, with $y_{0|t}^{(i)} \sim \mathcal{N}(y; y_t, \tilde{\sigma}_t^2 I)$ we have

$$\tilde{p}_t(y_t) \propto \int \exp(-\mathcal{E}(y))\mathcal{N}(y_t; y, \tilde{\sigma}_t^2 I)dy \tag{125}$$

$$\tilde{S}_t(y_t) = \nabla_{y_t} \log \tilde{p}_t(y_t) \approx \nabla_{y_t} \log \sum_{i=1}^K \exp(-\mathcal{E}(y_{0|t}^{(i)})) \tag{126}$$

$$\tilde{\mathcal{E}}_t(y_t) \approx -\log \frac{1}{K} \sum_{i=1}^K \exp(-\mathcal{E}(y_{0|t}^{(i)})) \tag{127}$$

Therefore, we can learn scores and energies of $y_t$ simply by following DEM and NEM for VE SDEs. Then for sampling, we can simulate the reverse SDE of $y_t$ and eventually, we have $x_0 = y_0$.

Instead, we can also learn energies and scores of $x_t$. By changing the variable, we can have

$$p_t(x_t) = \beta_t^{-1} \tilde{p}_t(\beta_t^{-1} x_t) = \beta_t^{-1} \tilde{p}_t(y_t) \tag{128}$$

$$S_t(x_t) = \beta_t^{-1} \tilde{S}_t(\beta_t^{-1} x_t) = \beta_t^{-1} \tilde{S}_t(y_t) \tag{129}$$

which provides us the energy and score estimator for $x_t$:

$$\mathcal{E}_t(x_t) \approx -\log \beta_t^{-1} \frac{1}{K} \sum_{i=1}^K \exp(-\mathcal{E}(x_{0|t}^{(i)})) \tag{130}$$

$$S_t(x_t) \approx \beta_t^{-1} \nabla_{x_t} \log \sum_{i=1}^K \exp(-\mathcal{E}(x_{0|t}^{(i)})) \tag{131}$$

$$x_{0|t}^{(i)} \sim \mathcal{N}(x; \beta_t^{-1} x_t, \tilde{\sigma}^2(t) I) \tag{132}$$

Typically, $\alpha$ is a non-negative function, resulting in $\beta(t)$ decreasing from 1 and can be close to 0 when $t$ is large. Therefore, the above equations realize that even though both the energies and scores for a general SDE can be estimated, the estimators are not reliable at large $t$ since $\beta_t^{-1}$ can be extremely large; while the SDE of $y_t$ (Eq. (124)) indicates that this equivalent VE SDE is scaled by $\beta_t^{-1}$, resulting that the variance of $y_t$ at large $t$ can be extremely large and requires much more MC samples for a reliable estimator. This issue can be a bottleneck of generalizing DEM, NEM, and BNEM to other SDE settings, therefore developing more reliable estimators for both scores and energies is of interest in future work.

## I.2 MC Energy Estimator as an Importance-Weighted Estimator

As for any SDEs, we can convert the modeling task to a VE process by changing variables, we stick to considering NEM with a VE process. Remember that the MC energy estimator aims to approximate the noised energy given by Eq. (6), which can be rewritten as:

$$\mathcal{E}_t(x_t) = -\log \int \exp(-\mathcal{E}(x))\mathcal{N}(x_t; x, \sigma_t^2 I)dx \tag{133}$$

$$= -\log \int \exp(-\mathcal{E}(x)) \frac{\mathcal{N}(x_t; x, \sigma_t^2 I)}{q_{0|t}(x|x_t)} q_{0|t}(x|x_t)dx \tag{134}$$

$$= -\log \mathbb{E}_{q_{0|t}(x|x_t)} \left[ \exp(-\mathcal{E}(x)) \frac{\mathcal{N}(x_t; x, \sigma_t^2 I)}{q_{0|t}(x|x_t)} \right] \tag{135}$$

The part inside the logarithm of Eq. (135) suggests an Importance Sampling technique for approximation, by using a proposal $q_{0|t}(x|x_t)$. Notice that when choosing a proposal symmetric to the perturbation kernal, *i.e.* $q_{0|t}(x|x_t) = \mathcal{N}(x; x_t, \sigma_t^2)$, Eq. (135) resembles the MC energy estimator we discussed in Sec. 3.2. Therefore, this formulation allows us to develop a better estimator by carefully selecting the proposal $q_{0|t}(x|x_t)$.

However, Owen (2013) shows that to minimize the variance of the IS estimator, the proposal $q_{0|t}(x|x_t)$ should be chosen roughly proportional to $f(x)\mu_{\text{target}}(x)$, where $f(x) = \exp(-\mathcal{E}(x))$ in our case. Finding such a proposal is challenging in high-dimension space or with a multimodal $\mu_{\text{target}}$. A potential remedy can be leveraging Annealed Importance Sampling (AIS; Neal (2001)).

## J  Memory-Efficient NEM

Differentiating the energy network to get denoising scores in (B)NEM raises additional computations compared with iDEM, which are usually twice the computation of forwarding a neural network and can introduce memory overhead due to saving the computational graph. The former issue can be simply solved by reducing the number of integration steps to half. To solve the latter memory issue, we propose a Memory-efficient NEM by revisiting Tweedie's formula (Efron, 2011). Given a VE noising process, $dx_t = g(t)dw_t$, where $w_t$ is Brownian motion and $\sigma_t^2 := \int_{s=0}^{t} g(s)^2 ds$, the Tweedie's formula can be written as:

$$\nabla \log p_t(x_t) = \frac{\mathbb{E}[x_0|x_t] - x_t}{\sigma_t^2} \tag{136}$$

$$\mathbb{E}[x_0|x_t] = \int x_0 p(x_0|x_t) dx_0 \tag{137}$$

$$= \int x_0 \frac{p(x_t|x_0)p_0(x_0)}{p_t(x_t)} dx_0 \tag{138}$$

By revisiting Eq. (5), it's noticable that the noised energy, $\mathcal{E}_t$ (Eq. (6)), shares the same partition function as $\mathcal{E}$, *i.e.* $\int \exp(-\mathcal{E}_t(x))dx = \int \exp(-\mathcal{E}(x))dx, \forall t \in [0, 1]$. Hence, Eq. (138) can be simplified as follows, which further suggests an MC estimator for the denoising score with no requirement for differentiation

$$\mathbb{E}[x_0|x_t] = \int x_0 \frac{\mathcal{N}(x_t; x_0, \sigma_t^2 I)\exp(-\mathcal{E}(x_0))}{\exp(-\mathcal{E}_t(x_t))} dx_0 \tag{139}$$

$$\approx \exp\left(-\left(\mathcal{E}(x_{0|t}) - \mathcal{E}_t(x_t)\right)\right) x_{0|t} \tag{140}$$

where $x_{0|t} \sim \mathcal{N}(x; x_t, \sigma_t^2 I)$. Given learned noised energy, we can approximate this denoiser estimator as follows:

$$D_\theta(x_t, t) := \exp\left(-\left(\mathcal{E}(x_{0|t}) - E_\theta(x_t, t)\right)\right) x_{0|t} \tag{141}$$

$$\tilde{D}_\theta(x_t, t) := \exp\left(-\left(E_\theta(x_{0|t}, 0) - E_\theta(x_t, t)\right)\right) x_{0|t} \tag{142}$$

where we can alternatively use $D_\theta$ or $\tilde{D}_\theta$ according to the accessability of clean energy $\mathcal{E}$, the relative computation between $\mathcal{E}(x)$ and $E_\theta(x, t)$, and the accuracy of $E_\theta(x, 0)$.

Experimental results of the memory-efficient NEM could be found in appendix M.7.

## K  TweeDEM: training denoising mean by Tweedie's formula

In this supplementary work, we propose TWEEDIE DEM (TweeDEM), by leveraging the Tweedie's formula (Efron, 2011) into DEM, *i.e.* $\nabla_x \log p_t(x) = \mathbb{E}_{p(x_0|x_t)}\left[(x_0 - x_t)/\sigma_t^2\right]$. TweeDEM is similar to the iEFM-VE proposed by Woo & Ahn (2024), which is a variant of iDEM corresponding to another family of generative model, flow matching. However iEFM targets the vector field, while TweeDEM targets the expected value of clean data given noisy data.

Table 2: Comparison between DEM, DDM, and TweeDEM.

| Score estimator: $\sum_i w_i s_i$, with $w_i = Softmax(\tilde{w})[i]$ | | | | |
|---|---|---|---|---|
| Sampler↓ Components→ | Weight Type | $\tilde{w}_i$ | Score Type | $s_i$ |
| DEM(Akhound-Sadegh et al., 2024) | System Energy | $\exp(-\mathcal{E}(x_{0|t}^{(i)}))$ | System Score | $-\nabla\mathcal{E}(x_{0|t}^{(i)})$ |
| DDM(Karras et al., 2022) | Gaussian Density | $\mathcal{N}(x_{0|t}^{(i)}; x_t, \sigma_t^2 I)$ | Gaussian Score | $\nabla\log\mathcal{N}(x_{0|t}^{(i)}; x_t, \sigma_t^2 I)$ |
| TweeDEM | System Energy | $\exp(-\mathcal{E}(x_{0|t}^{(i)}))$ | Gaussian Score | $\nabla\log\mathcal{N}(x_{0|t}^{(i)}; x_t, \sigma_t^2 I)$ |

We first derive an MC denoiser estimator, *i.e.* the expected clean data given a noised data $x_t$ at $t$

$$\mathbb{E}[x_0|x_t] = \int x_0 p(x_0|x_t) dx_0 \tag{143}$$

$$= \int x_0 \frac{q_t(x_t|x_0)p_0(x_0)}{p_t(x_t)} dx_0 \tag{144}$$

$$= \int x_0 \frac{\mathcal{N}(x_t; x_0, \sigma_t^2 I)\exp(-\mathcal{E}(x_0))}{\exp(-\mathcal{E}_t(x_t))} dx_0 \tag{145}$$

where the numerator can be estimated by an MC estimator $\mathbb{E}_{\mathcal{N}(x_t, \sigma_t^2 I)}[x\exp(-\mathcal{E}(x))]$ and the denominator can be estimated by another similar MC estimator $\mathbb{E}_{\mathcal{N}(x_t, \sigma_t^2 I)}[\exp(-\mathcal{E}(x))]$, suggesting we can approximate this denoiser through self-normalized importance sampling as follows

$$D_K(x_t, t) := \sum_{i=1}^{K} \frac{\exp(-\mathcal{E}(x_{0|t}^{(i)}))}{\sum_{j=1}^{K} \exp(-\mathcal{E}(x_{0|t}^{(j)}))} x_{0|t}^{(i)} \tag{146}$$

$$= \sum_{i=1}^{K} w_i x_{0|t}^{(i)} \tag{147}$$

where $x_{0|t}^{(i)} \sim \mathcal{N}(x_t, \sigma_t^2 I)$, $w_i$ are the importance weights and $D_K(x_t, t) \approx \mathbb{E}[x_0|x_t]$. Then a another MC score estimator can be constructed by plugging the denoiser estimator $D_K$ into Tweedie's formula, which resembles the one proposed by Huang et al. (2023)

$$\tilde{S}_K(x_t, t) := \sum_{i=1}^{K} w_i \frac{x_{0|t}^{(i)} - x_t}{\sigma_t^2} \tag{148}$$

where $\frac{x_{0|t}^{(i)} - x_t}{\sigma_t^2}$ resembles the vector fields $v_t(x_t)$ in Flow Matching. In another perspective, these vector fields can be seen as scores of Gaussian, *i.e.* $\nabla\log\mathcal{N}(x; x_t, \sigma_t^2 I)$, and therefore $\tilde{S}_K$ is an importance-weighted sum of Gaussian scores while $S_K$ can be expressed as an importance-weighted sum of system scores $-\nabla\mathcal{E}$. In addition, Karras et al. (2022) demonstrates that in Denoising Diffusion Models, the optimal scores are an importance-weighted sum of Gaussian scores, while these importance weights are given by the corresponding Gaussian density, *i.e.* $S_{\text{DM}}(x_t, t) = \sum_i \tilde{w}_i(x_{0|t}^{(i)} - x_t)/\sigma_t^2$ and $\tilde{w}_i \propto \mathcal{N}(x_{0|t}^{(i)}; x_t, \sigma_t^2 I)$. We summarize these three different score estimators in Table 2.

## L Experimental Details

### L.1 Energy functions

**GMM.** A Gaussian Mixture density in 2-dimensional space with 40 modes, which is proposed by Midgley et al. (2023). Each mode in this density is evenly weighted, with identical covariances,

$$\Sigma = \begin{pmatrix} 40 & 0 \\ 0 & 40 \end{pmatrix} \tag{149}$$

and the means $\{\mu_i\}_{i=1}^{40}$ are uniformly sampled from $[-40, 40]^2$, i.e.

$$p_{gmm}(x) = \frac{1}{40} \sum_{i=1}^{40} \mathcal{N}(x; \mu_i, \Sigma) \tag{150}$$

Then its energy is defined by the negative-log-likelihood, i.e.

$$\mathcal{E}^{GMM}(x) = -\log p_{gmm}(x) \tag{151}$$

For evaluation, we sample 1000 data from this GMM with *TORCH.RANDOM.SEED(0)* following Midgley et al. (2023); Akhound-Sadegh et al. (2024) as a test set.

**DW-4.** First introduced by Köhler et al. (2020), the DW-4 dataset describes a system with 4 particles in 2-dimensional space, resulting in a task with dimensionality $d = 8$. The energy of the system is given by the double-well potential based on pairwise Euclidean distances of the particles,

$$\mathcal{E}^{DW}(x) = \frac{1}{2\tau} \sum_{ij} a(d_{ij} - d_0) + b(d_{ij} - d_0)^2 + c(d_{ij} - d_0)^4 \tag{152}$$

where $a$, $b$, $c$ and $d_0$ are chosen design parameters of the system, $\tau$ the dimensionless temperature and $d_{ij} = \|x_i - x_j\|_2$ are Euclidean distance between two particles. Following Akhound-Sadegh et al. (2024), we set $a = 0$, $b = -4$, $c = 0.9$ $d_0 = 4$ and $\tau = 1$, and we use validation and test set from the MCMC samples in Klein et al. (2023a) as the "Ground truth" samples for evaluating.

**LJ-n**. This dataset describes a system consisting of $n$ particles in 3-dimensional space, resulting in a task with dimensionality $d = 3n$. Following Akhound-Sadegh et al. (2024), the energy of the system is given by $\mathcal{E}^{Tot}(x) = \mathcal{E}^{LJ}(x) + c\mathcal{E}^{osc}(x)$ with the Lennard-Jones potential

$$\mathcal{E}^{\mathrm{LJ}}(x) = \frac{\epsilon}{2\tau} \sum_{ij} \left( \left( \frac{r_m}{d_{ij}} \right)^6 - \left( \frac{r_m}{d_{ij}} \right)^{12} \right) \tag{153}$$

and the harmonic potential

$$\mathcal{E}^{osc}(x) = \frac{1}{2} \sum_i \|x_i - x_{COM}\|^2 \tag{154}$$

where $d_{ij} = \|x_i - x_j\|_2$ are Euclidean distance between two particles, $r_m$, $\tau$ and $\epsilon$ are physical constants, $x_{COM}$ refers to the center of mass of the system and $c$ the oscillator scale. We use $r_m = 1$, $\tau = 1$, $\epsilon = 1$ and $c = 0.5$ the same as Akhound-Sadegh et al. (2024). We test our models in LJ-13 and LJ-55, which correspond to $d = 65$ and $d = 165$ respectively. And we use the MCMC samples given by Klein et al. (2023a) as a test set.

## L.2 Evaluation Metrics

**2-Wasserstein distance** $\mathcal{W}_2$. Given empirical samples $\mu$ from the sampler and ground truth samples $\nu$, the 2-Wasserstein distance is defined as:

$$\mathcal{W}_2(\mu, \nu) = (\inf_\pi \int \pi(x, y) d^2(x, y) dx dy)^{\frac{1}{2}} \tag{155}$$

where $\pi$ is the transport plan with marginals constrained to $\mu$ and $\nu$ respectively. Following Akhound-Sadegh et al. (2024), we use the Hungarian algorithm as implemented in the Python optimal transport package (POT) (Flamary et al., 2021) to solve this optimization for discrete samples with the Euclidean distance $d(x, y) = \|x - y\|_2$. $\mathbf{x}\text{-}\mathcal{W}_2$ is based on the data and $\mathcal{E}\text{-}\mathcal{W}_2$ is based on the corresponding energy. For $n$-body systems, *i.e.* DW-4, LJ-13, and LJ-55, we take SE(3), *i.e.* rotational and translation equivariance into account. For given n-body point cloud pair $(X, Y)$, instead of using Euclidean distance, the distance is defined

as $d_{\text{Kabsch}}(X, Y) = \min_{R, t \in \text{SE(3)}} \|X - (YR^\top + t)\|_2$, where Kabsch algorithm is applied to find the optimal rotation and translation.

**Total Variation (TV)**. The total variation measures the dissimilarity between two probability distributions. It quantifies the maximum difference between the probabilities assigned to the same event by two distributions, thereby providing a sense of how distinguishable the distributions are. Given two distribution $P$ and $Q$, with densities $p$ and $q$, over the same sample space $\Omega$, the TV distance is defined as

$$TV(P, Q) = \frac{1}{2} \int_\Omega |p(x) - q(x)| dx \tag{156}$$

Following Akhound-Sadegh et al. (2024), for low-dimentional datasets like GMM, we use 200 bins in each dimension. For larger equivariant datasets, the total variation distance is computed over the distribution of the interatomic distances of the particles.

### L.3 Experiment Settings

We pin the number of reverse SDE integration steps for iDEM, NEM, BNEM and TweeDEM (see appendix K) as 1000 and the number of MC samples as 1000 in most experiments, except for the ablation studies.

**GMM-40.** For the basic model $f_\theta$, we use an MLP with sinusoidal and positional embeddings which has 3 layers of size 128 as well as positional embeddings of size 128. The replay buffer is set to a maximum length of 10000.

During training, the generated data was in the range $[-1, 1]$ so to calculate the energy it was scaled appropriately by unnormalizing by a factor of 50. Baseline models are trained with a geometric noise schedule with $\sigma_{\min} = 1e - 5$, $\sigma_{\max} = 1$; NEM and BNEM are trained with a cosine noise schedule with $\sigma_{\min} = 0.001$ and $\sigma_{\max} = 1$. We use $K = 500$ samples for computing the Bootstrap energy estimator $E_K^B$. We clip the norm of $S_K$, $s_\theta$ and $\nabla E_\theta$ to 70 during training and sampling. The variance controller for BNEM is set to be $\beta = 0.2$. All models are trained with a learning rate of $5e - 4$.

**DW-4.** All models use an EGNN with 3 message-passing layers and a 2-hidden layer MLP of size 128. All models are trained with a geometric noise schedule with $\sigma_{\min} = 1e - 5$, $\sigma_{\max} = 3$ and a learning rate of $1e - 3$ for computing $S_K$ and $E_K$. We use $K = 500$ samples for computing the Bootstrap energy estimator $E_K^B$. We clip the norm of $S_K$, $s_\theta$, and $\nabla E_\theta$ to 20 during training and sampling. The variance controller for BNEM is set to be $\beta = 0.2$.

**LJ-13.** All models use an EGNN with 5 hidden layers and hidden layer size 128. Baseline models are trained with a geometric noise schedule with $\sigma_{\min} = 0.01$ and $\sigma_{\max} = 2$; NEM and BNEM are trained with a geometric noise schedule with $\sigma_{\min} = 0.001$ and $\sigma_{\max} = 6.0$ to ensure the data well mixed to Gaussian. We use a learning rate of $1e - 3$, $K = 500$ samples for $E_K^B$, and we clipped $S_K$, $s_\theta$ and $\nabla E_\theta$ to a max norm of 20 during training and sampling. The variance controller for BNEM is set to be $\beta = 0.5$.

**LJ-55.** All models use an EGNN with 5 hidden layers and hidden layer size 128. All models are trained with a geometric noise schedule with $\sigma_{\min} = 0.5$ and $\sigma_{\max} = 4$. We use a learning rate of $1e - 3$, $K = 500$ samples for $E_K^B$. We clipped $S_K$ and $s_\theta$ to a max norm of 20 during training and sampling. And we clipped $\nabla E_\theta$ to a max norm of 1000 during sampling, as our model can capture better scores and therefore a small clipping norm can be harmful for sampling. The variance controller for BNEM is set to be $\beta = 0.4$.

**For all datasets.** We use clipped scores as targets for iDEM and TweeDEM training for all tasks. Meanwhile, we also clip scores during sampling in outer-loop of training, when calculating the reverse SDE integral. These settings are shown to be crucial especially when the energy landscape is non-smooth and exists extremely large energies or scores, like LJ-13 and LJ-55. In fact, targeting the clipped scores refers to learning scores of smoothed energies. While we're learning unadjusted energy for NEM and BNEM, the training can be unstable, and therefore we often tend to use a slightly larger $\sigma_{\min}$. Also, we smooth the Lennard-Jones potential through the cubic spline interpolation, according to Moore et al. (2024). Besides, we predict per-particle energies for DW-4 and LJ-n datasets, which can provide more information on the energy system. It shows that this setting can significantly stabilize training and boost performance.

Table 3: Ablation Study on applying energy smoothing based on Cubic Spline for iDEM

| Energy → | **LJ-55** ($d = 165$) | | |
|---|---|---|---|
| Sampler ↓ | $\mathbf{x}\text{-}\mathcal{W}_2\downarrow$ | $\mathcal{E}\text{-}\mathcal{W}_2\downarrow$ | **TV**$\downarrow$ |
| iDEM | $2.077 \pm 0.0238$ | $169347 \pm 2601160$ | $0.165 \pm 0.0146$ |
| iDEM (cubic spline smoothed) | $2.086 \pm 0.0703$ | $12472 \pm 8520$ | $0.142 \pm 0.0095$ |
| NEM (ours) | $\mathbf{1.898} \pm 0.0097$ | $\mathbf{118.57} \pm 106.62$ | $\mathbf{0.0991} \pm 0.0194$ |

## M   Supplementary Experients

### M.1   Explaination on baseline performance difference on LJ55 dataset

In our experiment, we observed a notable difference in performance on the LJ55 dataset compared to the results reported in previous work (Akhound-Sadegh et al., 2024). Upon examining the experimental setup, we found that the author of the previous study employed an additional round of Langevin dynamic optimization when sampling from the LJ55 target distribution. Specifically, the iDEM method only provides initial samples for subsequent optimization. However, this technique is not utilized with simpler distributions, which could lead to increased computational burdens when sampling from more complex distributions. And notably, these Langevin steps could be applied to samples generated by **any** samplers. To ensure a fair comparison, we did not conduct any post-optimization in our experiments.

### M.2   Comparing the Robustness of Energy-Matching and Score-Matching

In this section, we discussed the robustness of the energy-matching model(NEM) with the score-matching model(DEM) by analyzing the influence of the numbers of MC samples used for estimators and choice of noise schedule on the sampler's performance.

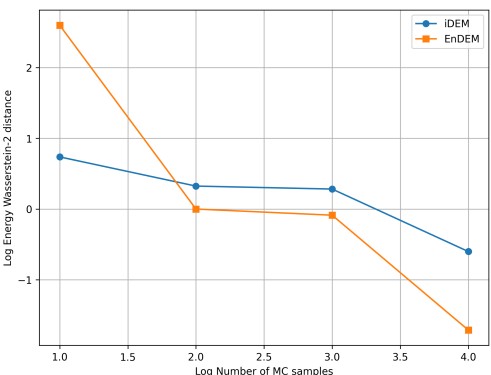

Figure 6: Comparison of the Energy Wasserstein-2 distance in DW4 benchmark between DEM and NEM across varying numbers of MC samples.

**Robustness with limited compute budget.** We first complete the robustness discussed in Section 5, by conducting experiments on a more complex benchmark - LJ-13.   reports different metrics of each sampler in different settings, *i.e.* 1000 integration steps and MC samples v.s. 100 integration steps and MC samples, and different tasks.

**Robustness v.s. Number of MC samples.** As in Fig. 6, NEM consistently outperforms iDEM when more than 100 MC samples are used for the estimator. Besides, NEM shows a faster decline when the number of MC samples increases. Therefore, we can conclude that the low variance of Energy-matching makes it more beneficial when we boost with more MC samples.

Table 4: Neural sampler performance comparison for 3 different energy functions. The number after the sampler, *e.g.* NEM-100, represents the number of integration steps and MC samples is 100. We measured the performance using data Wasserstein-2 distance(x-$\mathcal{W}_2$), Energy Wasserstein-2 distance($\mathcal{E}$-$\mathcal{W}_2$) and Total Variation(TV).

| Energy $\rightarrow$ | **GMM-40** ($d=2$) | | | **DW-4** ($d=8$) | | | **LJ-13** ($d=39$) | | |
|---|---|---|---|---|---|---|---|---|---|
| Sampler $\downarrow$ | **x**-$\mathcal{W}_2\downarrow$ | $\mathcal{E}$-$\mathcal{W}_2\downarrow$ | **TV**$\downarrow$ | **x**-$\mathcal{W}_2\downarrow$ | $\mathcal{E}$-$\mathcal{W}_2\downarrow$ | **TV**$\downarrow$ | **x**-$\mathcal{W}_2\downarrow$ | $\mathcal{E}$-$\mathcal{W}_2\downarrow$ | **TV**$\downarrow$ |
| iDEM-1000 | $4.21_{\pm0.86}$ | $1.63_{\pm0.61}$ | $0.81_{\pm0.03}$ | $\mathbf{0.42}_{\pm0.02}$ | $1.89_{\pm0.56}$ | $\mathbf{0.13}_{\pm0.01}$ | $0.87_{\pm0.00}$ | $77515.90_{\pm115028.07}$ | $0.06_{\pm0.01}$ |
| iDEM-100 | $8.21_{\pm5.43}$ | $60.49_{\pm70.12}$ | $0.82_{\pm0.03}$ | $0.50_{\pm0.03}$ | $2.80_{\pm1.72}$ | $0.16_{\pm0.01}$ | $0.88_{\pm0.00}$ | $1190.59_{\pm590}$ | $0.07_{\pm0.00}$ |
| NEM-1000 | $2.73_{\pm0.55}$ | $1.68_{\pm0.98}$ | $0.81_{\pm0.00}$ | $0.46_{\pm0.02}$ | $\mathbf{0.28}_{\pm0.08}$ | $0.28_{\pm0.13}$ | $0.02_{\pm0.01}$ | $5.01_{\pm2.56}$ | $\mathbf{0.03}_{\pm0.00}$ |
| NEM-100 | $5.28_{\pm0.89}$ | $44.56_{\pm39.56}$ | $0.91_{\pm0.02}$ | $0.48_{\pm0.02}$ | $0.85_{\pm0.52}$ | $0.14_{\pm0.01}$ | $0.88_{\pm0.00}$ | $13.14_{\pm225.45}$ | $0.04_{\pm0.00}$ |
| BNEM-1000 | $\mathbf{2.55}_{\pm0.47}$ | $\mathbf{0.36}_{\pm0.12}$ | $\mathbf{0.66}_{\pm0.08}$ | $0.49_{\pm0.01}$ | $\mathbf{0.29}_{\pm0.05}$ | $0.15_{\pm0.01}$ | $\mathbf{0.86}_{\pm0.00}$ | $\mathbf{0.62}_{\pm0.01}$ | $\mathbf{0.03}_{\pm0.00}$ |
| BNEM-100 | $3.66_{\pm0.30}$ | $1.87_{\pm1.00}$ | $0.79_{\pm0.04}$ | $0.49_{\pm0.02}$ | $0.38_{\pm0.09}$ | $0.14_{\pm0.01}$ | $0.87_{\pm0.00}$ | $5.93_{\pm3.01}$ | $\mathbf{0.03}_{\pm0.00}$ |

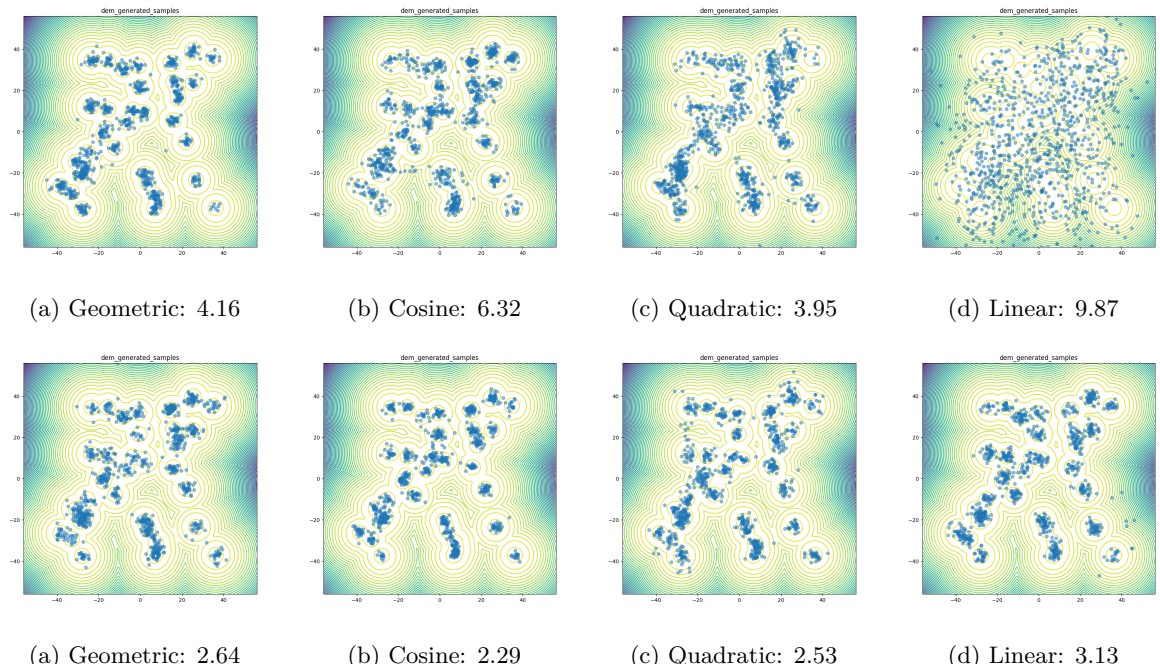

(a) Geometric: 4.16    (b) Cosine: 6.32    (c) Quadratic: 3.95    (d) Linear: 9.87

(a) Geometric: 2.64    (b) Cosine: 2.29    (c) Quadratic: 2.53    (d) Linear: 3.13

Figure 7: Comparison of different sampler (Above: iDEM; Below: NEM) when employing different noise schedules. The performances of x-$\mathcal{W}_2$ are listed.

**Robustness v.s. Different noise schedules.** Then, we evaluate the performance differences when applying various noise schedules. The following four schedules were tested in the experiment:

- **Geometric noise schedule**: The noise level decreases geometrically in this schedule. The noise at step $t$ is given by: $\sigma_t = \sigma_0^{1-t} \cdot \sigma_1^t$ where $\sigma_0 = 0.0001$ is the initial noise level, $\sigma_1 = 1$ is the maximum noise level, and $t$ is the time step.

- **Cosine noise schedule**: The noise level follows a cosine function over time, represented by: $\sigma_t = \sigma_1 \cdot \cos(\pi/2 \frac{1+\delta-t}{1+\delta})^2$, where $\delta = 0.008$ is a hyper-parameter that controls the decay rate.

- **Quadratic noise schedule**: The noise level follows a quadratic decay:$\sigma_t = \sigma_0 t^2$ where $\sigma_0$ is the initial noise level. This schedule applies a slow decay initially, followed by a more rapid reduction.

- **Linear noise schedule**: In this case, the noise decreases linearly over time, represented as: $\sigma_t = \sigma_1 t$

The experimental results are depicted in Fig. 7. It is pretty obvious that for iDEM the performance varied for different noise schedules. iDEM favors noise schedules that decay more rapidly to 0 when $t$ approaches 0. When applying the linear noise schedules, the samples are a lot more noisy than other schedules. This also proves our theoretical analysis that the variance would make the score network hard to train. On the contrary, all 4 schedules are able to perform well on NEM. This illustrates that the reduced variance makes NEM more robust and requires less hyperparameter tuning.

**Robustness in terms of Outliers.** Based on main experimental results, we set the maximum energy as (GMM-40: 100; DW-4: 0; LJ-13: 0; LJ-55: $-150$). We remove outliers based on these thresholds and recomputed the $\mathcal{E}\text{-}\mathcal{W}_2$. We report the new values as well as percentage of outliers in Table 5, which shows that the order of performance (BNEM>NEM>iDEM) still holds in terms of better $\mathcal{E}\text{-}\mathcal{W}_2$ value and less percentage of outliers.

Table 5: $\mathcal{E}\text{-}\mathcal{W}_2$ w/o outliers (outlier%) for different models and datasets. **Bold** indicates the best value and underline indicates the second one.

| Sampler↓ Energy→ | GMM-40 ($d=2$) | DW-4 ($d=8$) | LJ-13 ($d=39$) | LJ-55 ($d=165$) |
|---|---|---|---|---|
| iDEM | 0.138 (0.0%) | 8.658 (0.02%) | 88.794 (4.353%) | 21255 (0.29%) |
| NEM (ours) | 0.069 (0.0%) | 4.715 (**0.0%**) | 5.278 (0.119%) | 98.206 (0.020%) |
| BNEM (ours) | **0.032** (0.0%) | **1.050** (**0.0%**) | **1.241** (**0.025%**) | **11.401** (**0.0%**) |

## M.3 Empirical Analysis of the Variance of $E_K$ and $S_K$

To justify the theoretical results for the variance of the MC energy estimator (Eq. (9)) and MC score estimator (Eq. (7)), we first empirically explore a 2D GMM. For better visualization, the GMM is set to be evenly weighted by 10 modes located in $[-1, 1]^2$ with identical variance $1/40$ for each component, resulting in the following density

$$p'_{GMM}(x) = \frac{1}{10} \sum_{i=1}^{10} \mathcal{N}\left(x; \mu_i, \frac{1}{40} I\right) \tag{157}$$

while the marginal perturbed distribution at $t$ can be analytically derived from Gaussian's property:

$$p_t(x) = (p'_{GMM} * \mathcal{N}(0, \sigma_t^2))(x_t) = \frac{1}{10} \sum_{i=1}^{10} \mathcal{N}\left(x; \mu_i, \left(\frac{1}{40} + \sigma_t^2\right) I\right) \tag{158}$$

given a VE noising process.

We empirically estimate the variance for each pair of $(x_t, t)$ by simulating 10 times the MC estimators. Besides, we estimate the expected variance over $x$ for each time $t$, i.e. $\mathbb{E}_{p_t(x_t)}[\text{Var}(E_K(x_t, t))]$ and $\mathbb{E}_{p_t(x_t)}[\text{Var}(S_K(x_t, t))]$.

Fig. 8a shows that, the variance of both MC energy estimator and MC score estimator increase as time increases. In contrast, the variance of $E_K$ can be smaller than that of $S_K$ in most areas, especially when the energies are low (see Fig. 8c), aligning our Proposition 3.3. Fig. 8b shows that in expectation over true data distribution, the variance of $E_K$ is always smaller than that of $S_K$ across $t \in [0, 1]$.

## M.4 Empirical Analysis of the Bias of Bootstrapping

To show the improvement gained by bootstrapping, we deliver an empirical study on the GMM-40 energy in this section. As illustrated in appendix L.3, the modes of GMM-40 are located between $[-40, 40]^2$ with small variance. Therefore, the sub-Gaussianess assumption is natural. According to appendix C and appendix D, the analytical bias of the MC energy estimator (Eq. (9)) and Bootstrapped energy estimator (Eq. (17)) can

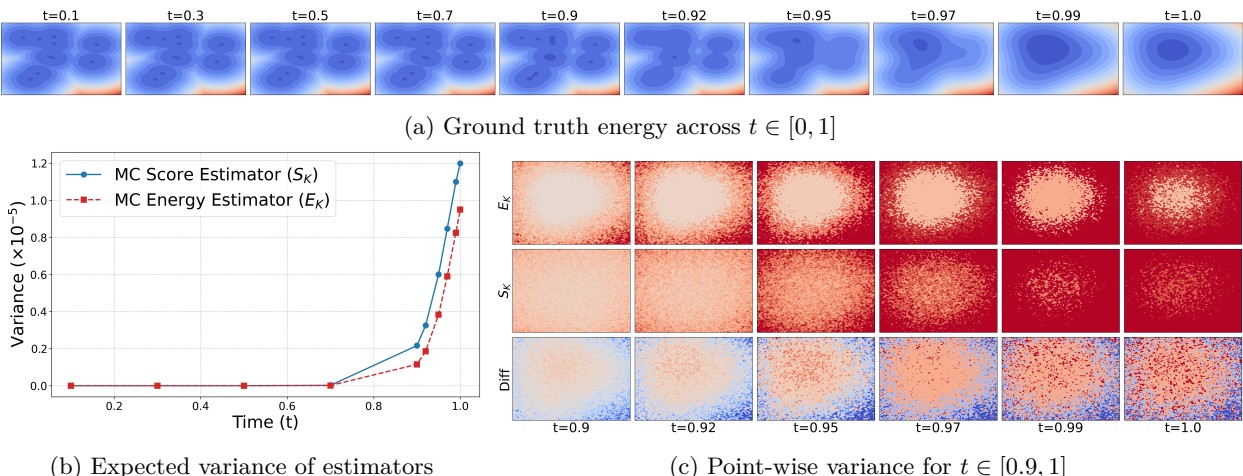

(a) Ground truth energy across $t \in [0, 1]$

(b) Expected variance of estimators

(c) Point-wise variance for $t \in [0.9, 1]$

Figure 8: (a) the ground truth energy of the target GMM from $t = 0$ to $t = 1$; (b) the estimation of expected variance of $x$ from $t = 0$ to $t = 1$, computed by a weighted sum over the variance of estimator at each location with weights equal to the marginal density $p_t$; (c) the variance of MC score estimator and MC energy estimator, and their difference (Var[score]-Var[energy]) for $t$ from 0.9 to 1, we ignore the plots from $t \in [0, 0.9]$ since the variance of both estimators are small. The colormap ranges from blue (low) to red (high), where blues are negative and reds are positive.

be computed by Eq. (25) and Eq. (18), respectively. We provide these two bias terms here for reference,

$$\text{Bias}(E_K(x_t, t)) = \frac{v_{0t}(x_t)}{2m_t(x_t)^2 K} \tag{159}$$

$$\text{Bias}(E_K(x_t, t, s; \theta)) = \frac{v_{0t}(x_t)}{2m_t^2(x_t) K^{n+1}} + \sum_{j=1}^{n} \frac{v_{0s_j}(x_t)}{2m_{s_j}^2(x_t) K^j} \tag{160}$$

Given a Mixture of Gaussian with $K$ components, $p_0(x) = \sum_k \pi_k \mathcal{N}(x; \mu_k, \Sigma_k)$ and $\mathcal{E}(x) = -\log p_0(x)$, $m_t(x)$ and $v_{0t}$ can be calculated as follows:

$$m_t(x_t) = \exp(-\mathcal{E}_t(x_t)) \tag{161}$$

$$= \int \mathcal{N}(x_t; x, \sigma_t^2 I) \exp(-\mathcal{E}(x)) dx \tag{162}$$

$$= \int \mathcal{N}(x_t; x, \sigma_t^2 I) p_0(x) dx \tag{163}$$

$$= \sum_k \pi_k \int \mathcal{N}(x_t; x, \sigma_t^2 I) \mathcal{N}(x; \mu_k, \Sigma_k) dx \tag{164}$$

$$= \sum_k \pi_k \mathcal{N}(x_t; \mu_k, \sigma_t^2 I + \Sigma_k) \tag{165}$$

and

$$v_{0t}(x_t) = \text{Var}_{\mathcal{N}(x;x_t,\sigma_t^2 I)}(\exp(-\mathcal{E}(X))) \tag{166}$$

$$= \int \mathcal{N}(x_t; x, \sigma_t^2 I)p_0(x)^2 dx - m_t^2(x_t) \tag{167}$$

$$= \sum_{j,k} \pi_j \pi_k \int \mathcal{N}(x_t; x, \sigma_t^2 I)\mathcal{N}(x; \mu_k, \Sigma_k)\mathcal{N}(x; \mu_j, \Sigma_j)dx - m_t^2(x_t) \tag{168}$$

$$= \sum_{j,k} \pi_j \pi_k \int \mathcal{N}(x_t; x, \sigma_t^2 I)\mathcal{N}(x; \mu_{jk}, \Sigma_{jk})C_{jk}\frac{|\Sigma_{jk}|^{1/2}}{(2\pi)^{d/2}|\Sigma_j|^{1/2}|\Sigma_k|^{1/2}}dx - m_t^2(x_t) \tag{169}$$

$$= \sum_{j,k} \pi_j \pi_k C_{jk}\frac{|\Sigma_{jk}|^{1/2}}{(2\pi)^{d/2}|\Sigma_j|^{1/2}|\Sigma_k|^{1/2}}\mathcal{N}(x_t; \mu_{jk}, \sigma_t^2 I + \Sigma_{jk}) - m_t^2(x_t), \tag{170}$$

where

$$\Sigma_{jk} = (\Sigma_j^{-1} + \Sigma_k^{-1})^{-1} \tag{171}$$

$$\mu_{jk} = \Sigma_{jk}(\Sigma_j^{-1}\mu_j + \Sigma_k^{-1}\mu_k) \tag{172}$$

$$C_{jk} = \exp\left(-\frac{1}{2}\left[\mu_j^\top \Sigma_j^{-1}\mu_j + \mu_k^\top \Sigma_k^{-1}\mu_k - (\Sigma_j^{-1}\mu_j + \Sigma_k^{-1}\mu_k)^\top \Sigma_{jk}(\Sigma_j^{-1}\mu_j + \Sigma_k^{-1}\mu_k)\right]\right) \tag{173}$$

In our GMM-40 case, the covariance for each component are identical and diagonal, *i.e.* $\Sigma_k \equiv \Sigma = vI$. By plugging it into the equations, we can simplify the $m_t(x_t)$ and $v_{0t}(x_t)$ terms as follows

$$m_t(x_t) = \sum_k \frac{1}{K}\mathcal{N}(x_t; \mu_k, (\sigma_t^2 + v)I) \tag{174}$$

$$v_{0t}(x_t) = \sum_{j,k} \frac{1}{K^2}\frac{\exp\left(-\frac{1}{4v}(\mu_j - \mu_k)^\top(\mu_j - \mu_k)\right)}{\sqrt{2}(2\pi v)}\mathcal{N}\left(x_t; \frac{1}{2}(\mu_j + \mu_k), (\sigma_t^2 + v/2)I\right) - m_t^2(x_t) \tag{175}$$

We computed the analytical bias terms and visualize in Fig. 9. Fig. 9a visualizes the bias of the both NEM and BNEM over the entire space. It shows that (1) bootstrapped energy estimator can have less bias (contributed by bias of EK and variance of training target); (2) If $E_K$ is already bias, i.e. the "red" regions in the first row of Fig. 9a when $t = 0.1$, bootstrapping can not gain any improvement, which is reasonable; (3) However, if $E_K$ has low bias, i.e. the "blue" regions when $t = 0.1$, the Bootstrapped energy estimator can result in lower bias estimation, superioring MC energy estimator; (4) In low energy region, both MC energy estimator and Bootstrapped one result in accurate estimation. However, in a bi-level iterated training fashion, we always probably explore high energy at the beginning. Therefore, due to the less biasedness of Bootstrapped estimator at high energy regions, we're more likely to have more informative pseudo data which can further improve the model iteratively.

On the other hand, we ablate different settings of Num. of MC samples and the variance-control (VC) parameter. We visualize the results in Fig. 9b. The results show that, with proper VC, bootstrapping allows us to reduce the bias with less MC samples, which is desirable in high-dimensional and more complex problems.

## M.5 Complexity Analysis

To compare the time complexity between iDEM, NEM and BNEM, we let: (1) In the outer-loop of training, we have $T$ integration steps and batch size $B$; (2) In the inner-loop, we have $L$ epochs and batch size $B$. Let $\Gamma_{\text{NN}}(B)$ be the time complexity of evaluating a neural network w.r.t. $B$ data points, $\Gamma_{\mathcal{E}}(B)$ be the time complexity of evaluating the clean energy w.r.t. $B$ data points, and $K$ be the number of MC samples used. Since differentiating a function $f$ using the chain rule requires approximately twice the computation as evaluating $f$, we summarise the time complexity of iDEM, NEM, and BNEM in Table 6. It shows that in principle, in the inner-loop, NEM can be slightly faster than iDEM, while BNEM depends on the relativity between complexity of evaluating the neural network and evaluating the clean energy function.

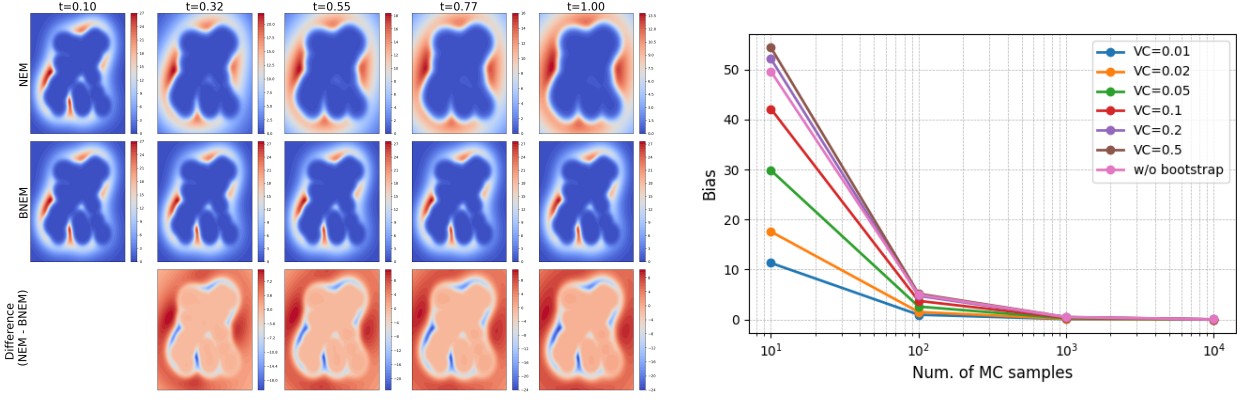

(a) Bias with VC=0.1

(b) Bias with different VC and Num. MC samples.

Figure 9: Empirical analysis on bias with bootstrapping, on GMM-40.

Table 6: Time complexity of different neural samplers.

| Sampler↓ Phase→ | Inner-loop | Outer-loop (or Sampling) |
|---|---|---|
| iDEM | $\mathcal{O}\left(L(2K\Gamma_{\mathcal{E}}(B) + 2\Gamma_{\mathrm{NN}}(B))\right)$ | $\mathcal{O}\left(T\Gamma_{\mathrm{NN}}(B)\right)$ |
| NEM (ours) | $\mathcal{O}\left(L(K\Gamma_{\mathcal{E}}(B) + 2\Gamma_{\mathrm{NN}}(B))\right)$ | $\mathcal{O}\left(2T\Gamma_{\mathrm{NN}}(B)\right)$ |
| BNEM (ours) | $\mathcal{O}\left(L(K\Gamma_{\mathrm{NN}}(B) + 2\Gamma_{\mathrm{NN}}(B))\right)$ | $\mathcal{O}\left(2T\Gamma_{\mathrm{NN}}(B)\right)$ |

Table 7: Time comparison (in seconds) of different samplers for both inner-loop (network training) and outer-loop (denoising sampling) across different energies

| Energy → | **GMM-40** ($d=2$) | | **DW-4** ($d=8$) | | **LJ-13** ($d=39$) | | **LJ-55** ($d=165$) | |
|---|---|---|---|---|---|---|---|---|
| Sampler ↓ | **Inner-loop** | **Outer-loop** | **Inner-loop** | **Outer-loop** | **Inner-loop** | **Outer-loop** | **Inner-loop** | **Outer-loop** |
| iDEM | 1.657 | 1.159 | 6.783 | 2.421 | 21.857 | 21.994 | 36.158 | 47.477 |
| NEM (ours) | 1.658 | 2.252 | 5.217 | 8.517 | 14.646 | 52.563 | 17.171 | 114.601 |
| BNEM (ours) | 1.141 | 2.304 | 25.552 | 7.547 | 68.217 | 52.396 | 113.641 | 115.640 |

Table 8: Memory comparison (in GiB) of different samplers during sampling across different energies

| Sampler ↓ Energy → | **GMM-40** ($d=2$) | **DW-4** ($d=8$) | **LJ-13** ($d=39$) | **LJ-55** ($d=165$) |
|---|---|---|---|---|
| iDEM | 0.02 | 0.06 | 0.08 | 1.30 |
| NEM/BNEM (ours) | 0.02 | 0.11 | 0.25 | 4.30 |

Table 7 reports the time usage per inner-loop and outer-loop. It shows that due to the need for differentiation, the sampling time, *i.e.* outer-loop, of BNEM/NEM is approximately twice that of iDEM. In contrast, the inner-loop time of NEM is slightly faster than that of iDEM, matching the theoretical time complexity, and the difference becomes more pronounced for more complex systems such as LJ-13 and LJ-55. For BNEM, the sampling time is comparable to NEM, but the inner-loop time depends on the relative complexity of evaluating the clean energy function versus the neural network, which can be relatively higher.

Table 8 reports the GPU memory overhead of iDEM, NEM, and BNEM, where the network architectures are detailed in appendix L. The batch size for calculation are 1024, 1024, 128, and 128, respectively.

Table 9: Comparison between iDEM, NEM, and BNEM, with similar computational budget.

| Energy → | **LJ-13** ($d = 39$) | | | **LJ-55** ($d = 165$) | | |
|---|---|---|---|---|---|---|
| Sampler ↓ | x-$\mathcal{W}_2$↓ | $\mathcal{E}$-$\mathcal{W}_2$↓ | **TV**↓ | x-$\mathcal{W}_2$↓ | $\mathcal{E}$-$\mathcal{W}_2$↓ | **TV**↓ |
| iDEM | 0.870 | 6670 | 0.0600 | 2.060 | 17651 | 0.160 |
| NEM-500 (ours) | 0.870 | 31.877 | 0.0377 | 1.896 | 11018 | 0.0955 |
| BNEM-500 (ours) | 0.866 | 2.242 | 0.0329 | 1.890 | 25277 | 0.113 |

Table 10: Performance comparison between NEM and ME-NEM.

| Energy → | **GMM-40** ($d = 2$) | | | **DW-4** ($d = 8$) | | | **LJ-13** ($d = 39$) | | |
|---|---|---|---|---|---|---|---|---|---|
| Sampler ↓ | x-$\mathcal{W}_2$↓ | $\mathcal{E}$-$\mathcal{W}_2$↓ | **TV**↓ | x-$\mathcal{W}_2$↓ | $\mathcal{E}$-$\mathcal{W}_2$↓ | **TV**↓ | x-$\mathcal{W}_2$↓ | $\mathcal{E}$-$\mathcal{W}_2$↓ | **TV**↓ |
| NEM (ours) | 1.808 | 0.846 | 0.838 | 0.479 | 2.956 | 0.14 | 0.866 | 27.362 | 0.0369 |
| ME-NEM (ours) | 2.431 | 0.107 | 0.813 | 0.514 | 1.649 | 0.164 | 0.88 | 20.161 | 0.0338 |

## M.6 Performance Gain Under the Same Computational Budget

It's noticable that computing the scores by differentiating the energy network outputs, *i.e.* $\nabla_{x_t} E_\theta(x_t, t)$, requires twice of computation compared with iDEM which computes $s_\theta(x_t, t)$ by one neural network evaluation. In this section, we limit the computational budget during sampling of both NEM and BNEM by reducing their integration steps to half. We conduct experiment on LJ-13 and LJ-55, where we reduce the reverse SDE integration steps in both NEM and BNEM from 1000 to 500. Metrics are reported in Table 9. It shows that with similar computation, NEM and BNEM can still outperform iDEM.

## M.7 Experiments for Memory-Efficient NEM

In this section, we conduct experiments on the proposed Memory-Efficient NEM (ME-NEM). The number of integration steps and MC samples are all set to 1000. ME-NEM is proposed to reduce the memory overhead caused by differentiating the energy network in NEM, which leverages the Tweedie's formula to establish a 1-sample MC estimator for the denoising score. In principle, ME-NEM doesn't require neural network differentiation, avoiding saving the computational graph. Though it still requires evaluating the neural network twice (see Eq. (142)), this only requires double memory usage of iDEM and can be computed parallelly, resulting in a similar speed of sampling with iDEM. In this section, we simply show a proof-of-concept experiment on ME-NEM, while leaving more detailed experiments as our future work.

Table 10 reports the performance of NEM and ME-NEM, showcasing that ME-NEM can achieve similar results even though it leverages another MC estimator during sampling.

## M.8 Experiments for TweeDEM

In appendix K, we propose TweeDEM, a variant of DEM by leveraging Tweedie's formula (Efron, 2011), which theoretically links iDEM and iEFM-VE and suggests that we can simply replace the score estimator $S_K$ (Eq. (7)) with $\tilde{S}_K$ (Eq. (148)) to reconstruct a iEFM-VE. We conduct experiments for this variant with the aforementioned GMM and DW-4 potential functions.

**Setting.** We follow the ones aforementioned, but setting the steps for reverse SDE integration 1000, the number of MC samples 500 for GMM and 1000 for DW-4. We set a quadratic noise schedule ranging from 0 to 3 for TweeDEM in DW-4.

To compare the two score estimators $S_K$ and $\tilde{S}_K$ fundamentally, we first conduct experiments using these ground truth estimations for reverse SDE integration, *i.e.* samplers without learning. In addition, we consider using a neural network to approximate these estimators, *i.e.* iDEM and TweeDEM.

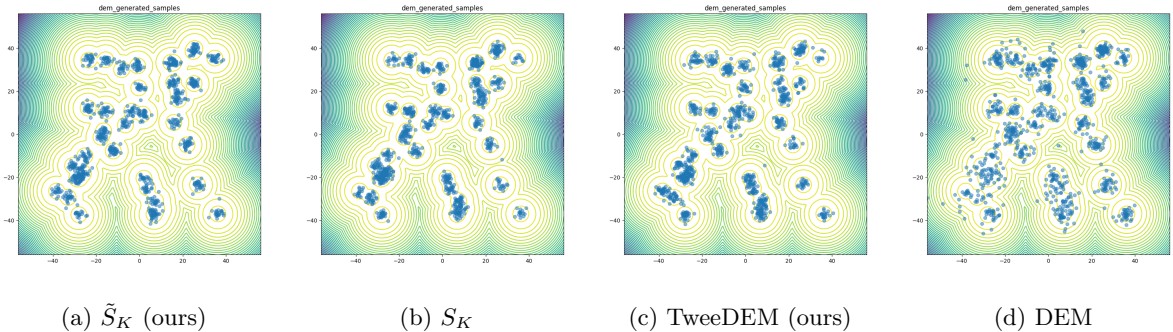

(a) $\tilde{S}_K$ (ours)          (b) $S_K$          (c) TweeDEM (ours)          (d) DEM

Figure 10: Sampled points from samplers applied to GMM-40 potentials, with the ground truth represented by contour lines. $\tilde{S}_K$ and $S_K$ represent using these ground truth estimators for reverse SDE integration.

Table 11: Sampler performance comparison for GMM-40 and DW-4 energy function. we measured the performance using data Wasserstein-2 distance(x-$\mathcal{W}_2$), Energy Wasserstein-2 distance($\mathcal{E}$-$\mathcal{W}_2$), and Total Variation(TV). † We compare the optimal number reported by Woo & Ahn (2024) and Akhound-Sadegh et al. (2024). . - indicates metric non-reported.

| Energy → | **GMM-40** ($d=2$) | | | **DW-4** ($d=8$) | | |
|---|---|---|---|---|---|---|
| Sampler ↓ | x-$\mathcal{W}_2\downarrow$ | $\mathcal{E}$-$\mathcal{W}_2\downarrow$ | **TV**↓ | x-$\mathcal{W}_2\downarrow$ | $\mathcal{E}$-$\mathcal{W}_2\downarrow$ | **TV**↓ |
| $S_K$ | 2.864 | **0.010** | **0.812** | 1.841 | **0.040** | 0.092 |
| $\tilde{S}_K$ (ours) | **2.506** | 0.124 | 0.826 | **1.835** | 0.145 | **0.087** |
| iDEM† | 3.98 | - | 0.81 | 2.09 | - | 0.09 |
| iDEM (rerun) | 6.406 | 46.90 | 0.859 | **1.862** | **0.030** | **0.093** |
| iEFM-VE† | 4.31 | - | - | 2.21 | - | - |
| iEFM-OT† | 4.21 | - | - | 2.07 | - | - |
| TweeDEM (ours) | **3.182** | **1.753** | **0.815** | 1.910 | 0.217 | 0.120 |

Table 11 reports x-$\mathcal{W}_2$, $\mathcal{E}-\mathcal{W}_2$, and TV for GMM and DW-4 potentials. Table 11 shows that when using the ground truth estimators for sampling, there's no significant evidence demonstrating the privilege between $S_K$ and $\tilde{S}_K$. However, when training a neural sampler, TweeDEM can significantly outperform iDEM (rerun), iEFM-VE, and iEFM-OT for GMM potential. While for DW4, TweeDEM outperforms iEFM-OT and iEFM-VE in terms of $x - \mathcal{W}_2$ but are not as good as our rerun iDEM.

Fig. 10 visualizes the generated samples from ground truth samplers, *i.e.* $S_K$ and $\tilde{S}_K$, and neural samplers, *i.e.* TweeDEM and iDEM. It shows that the ground truth samplers can generate well mode-concentrated samples, as well as TweeDEM, while samples generated by iDEM are not concentrated on the modes and therefore result in the high value of $\mathcal{W}_2$ based metrics. Also, this phenomenon aligns with the one reported by Woo & Ahn (2024), where the iEFM-OT and iEFM-VE can generate samples more concentrated on the modes than iDEM.

Above all, simply replacing the score estimator $S_K$ with $\tilde{S}_K$ can improve generated data quality and outperform iEFM in GMM and DW-4 potentials. Though TweeDEM can outperform the previous state-of-the-art sampler iDEM on GMM, it is still not as capable as iDEM on DW-4. Except scaling up and conducting experiments on larger datasets like LJ-13, combing $S_K$ and $\tilde{S}_K$ is of interest in the future, which balances the system scores and Gaussian ones and can possibly provide more useful and less noisy training signals. In addition, we are considering implementing a denoiser network for TweeDEM as our future work, which might stabilize the training process.

Table 12: Experimental results for GMM and DW-4 compared with Hamiltonian Monte Carlo (HMC) and Parallel Tempering (PT). The number of evaluations of the energy function is indicated in parentheses. HMC and PT evaluate the true target energy, while NEM and BNEM use learned energy functions.

| Energy $\rightarrow$ | **GMM-40** ($d = 2$) | | | **DW-4** ($d = 8$) | | |
|---|---|---|---|---|---|---|
| Sampler $\downarrow$ | $\mathbf{x}\text{-}\mathcal{W}_2\downarrow$ | $\mathcal{E}\text{-}\mathcal{W}_2\downarrow$ | **TV**$\downarrow$ | $\mathbf{x}\text{-}\mathcal{W}_2\downarrow$ | $\mathcal{E}\text{-}\mathcal{W}_2\downarrow$ | **TV**$\downarrow$ |
| HMC(100) | 12.40 | 5.73 | 0.87 | 0.56 | 455.00 | 0.45 |
| HMC(1000) | 11.26 | 0.03 | 0.85 | 0.42 | 3.44 | 0.11 |
| PT(100) | 9.02 | 1945.67 | 0.79 | 0.47 | 38.53 | 0.20 |
| PT(1000) | 5.24 | 2.17 | 0.85 | 0.49 | 3.70 | 0.10 |
| NEM(100) | 5.28 | 44.56 | 0.91 | 0.48 | 0.85 | 0.14 |
| BNEM(100) | 3.66 | 1.87 | 0.79 | 0.49 | 0.38 | 0.14 |

## M.9 Experimental results on non-neural samplers

As shown in Table 12, we compared our methods with traditional samplers that necessitate direct evaluation of the target energy function during sampling. We conducted a hyperparameter sweep on the step sizes for both methods based on $\mathcal{E}\text{-}\mathcal{W}_2$. The results presented in Table 12 represent the mean performance from three runs with varying random seeds.

In the GMM-40 experiments, Hamiltonian Monte Carlo (HMC) demonstrates impressive performance in sampling low-energy data points. However, it struggles to accurately weigh the different modes that yield high $x\text{-}\mathcal{W}_2$. Merely increasing the number of sampling steps does not address this issue, indicating that HMC often becomes trapped in local minima when dealing with sharply defined target distributions. On the other hand, introducing more randomness into the sampling process significantly increases the number of outliers. For Parallel Tempering (PT), increasing the number of sampling steps results in lower energy values and a more accurate capture of the modes of the target distribution. However, even with ten times the sampling steps of BNEM, it still underperforms compared to our best neural sampler. In the DW-4 experiments, both PT and HMC demonstrate good performance on $x\text{-}\mathcal{W}_2$; however, they still produce a significant number of outliers, which contribute to high energy.

## Assets and Licenses

**Datasets**

- DW-4, LJ-13, and LJ-55 (Klein et al., 2023b): MIT license

**Codebases**

- DW-4, LJ-13, and LJ-55 target energy functions: MIT license (https://github.com/noegroup/bgflow)

- FAB (Midgley et al., 2023): MIT license (https://github.com/lollcat/fab-torch)

- iDEM (Akhound-Sadegh et al., 2024): MIT license (https://github.com/jarridrb/DEM)

