# OpenReview forum: "BNEM: A Boltzmann Sampler Based on Bootstrapped Noised Energy Matching"
_TMLR — Accepted by TMLR_

### Review · Reviewer_6wXU · 2025-12-08

**Summary Of Contributions:**

This paper introduces Noised Energy Matching (NEM), a diffusion‐based sampler that learns noised energy functions rather than noised scores, and Bootstrap NEM (BNEM), which further reduces variance by bootstrapping high-noise energy estimates from lower-noise estimates. The authors present theoretical analyses demonstrating lower variance for energy-based Monte Carlo estimators, provide a bias–variance characterization of the bootstrap mechanism, and show strong empirical results on several Boltzmann sampling tasks, including Lennard–Jones systems with up to 55 particles.

**Audience:**

Yes

**Audience Explanation:**

NEM and BNEM introduce conceptually simple yet impactful improvements to diffusion-based Boltzmann samplers, supported by both theory and experiments. Researchers working on diffusion models, statistical mechanics, and sampling algorithms will likely find the insights especially relevant.

**Claims And Evidence:**

No

**Claims Explanation:**

Mostly yes, with a few areas where additional evidence would strengthen the work.

1. Bias introduced by BNEM is not directly quantified. The theory shows accumulated bias, but the experiments only evaluate final sampling quality, not the bias of the learned energies. This weakens the empirical support for claims about bias–variance tradeoffs.

2. Memory and computational overhead of differentiating the energy network during sampling is not evaluated. Since the paper explicitly states this as a limitation, evidence comparing memory or runtime relative to iDEM would strengthen the claims of efficiency.

3. The rejection scheme for bootstrapping is heuristic and lacks empirical justification. Although BNEM works well in practice, ablations or sensitivity studies would make the claim of its effectiveness more convincing.

**Requested Changes:**

1. Report memory and computational overhead during sampling.
2. A short discussion of the theoretical motivation of the rejection scheme

---

### Review · Reviewer_d83M · 2025-12-11

**Summary Of Contributions:**

This paper proposes Noised Energy Matching (NEM) and a bootstrapped variant, i.e., bootstrapped NEM (BNEM), that aim to generate independent samples from a Boltzmann distribution.  Specifically, they work in a setting with access only to an energy function, without samples, and propose an approach that utilizes diffusion-based neural samplers.  Compared with other methods, e.g., Iterated Denoising Energy Matching (iDEM), this work aims to learn noised energy functions rather than score functions.  The authors show that in ideal cases (infinite data, time, and model capacity), NEM achieves the same score as iDEM.  In addition, in practical settings, they theoretically justify reduced variance over iDEM.   Empirically, the proposed methods consistently outperform existing baselines.

I am not an expert in diffusion, generative models, or energy-based methods.  However, I found the paper to be reasonably clear and accessible. The authors do a good job motivating the problem, explaining the high-level ideas behind their methods, and structuring the theoretical and empirical results in a way that a non-specialist can follow.  For that reason, I'd say the paper is generally well written and clear.  Beyond this, I am not familiar enough with the area to provide a substantive assessment of its strengths and weaknesses.

**Audience:**

Yes

**Audience Explanation:**

Generative models are of significant interest to a large portion of the ML community, so this paper would easily be of interest to the TMLR community.

**Broader Impact Concerns:**

None.

**Claims And Evidence:**

Yes

**Claims Explanation:**

The empirical claims in the paper are supported by clear numerical experiments comparing the proposed methods with relevant baselines, which are consistent with the authors’ stated contributions. My evaluation is based primarily on the empirical and methodological presentation, as I am not a domain expert and therefore did not evaluate the correctness of the detailed theory.

**Requested Changes:**

I have no specific revision requests.

---

### Review · Reviewer_Rcqo · 2025-12-26

**Summary Of Contributions:**

The paper proposes NEM that reduces the variance and BNEM which uses a bootstrapping target to further reduce the variance at the cost of increasing the bias. Both approaches learn the energy function as opposed to iDEM that learns the score function instead. A strength is that the approach indeed works with reduced variance that is quite strongly validated by experiments. However, it is not supported well by the theory and math that come with confusing notations and the following questions.

**Audience:**

Yes

**Audience Explanation:**

It is based on the diffusion methods that are actively studied in ML; the Boltzman sampler would be an interesting topic in probabilistic ML.

**Broader Impact Concerns:**

N / A.

**Claims And Evidence:**

No

**Claims Explanation:**

As explained above, it seems that the claims are not supported well by the theory for which I addressed the issues below. In the review, I focused on the technical aspects especially the math in the body and the proofs.

**Requested Changes:**

# Main Concerns

* **Confusing Notations.** The authors introduced in Eqn. (5) the Gaussian distribution/pdf $\mathcal{N}(x; x_t, \sigma_t I) \equiv \mathcal{N}(x_t; x_0, \sigma_t I)$ and used it thereafter. I understand that they are equivalent as $x = x_0$ implicitly and $\|x_t - x_0\|^2 = \|x_0 - x_t\|^2$. However, it is still confusing in the subsequent formula. For example,
   * $p_t(x_t)$ in Eqn. (5) is given w.r.t. $x_t$ but $\mathcal{E}_t(x)$ in Eqn. (6) is given w.r.t. $x$ which is the integration variable in Eqn. (5). The expression (6) is rewritten as $\mathcal{E}_t(x) = - \log p_t(x)$, but what is the difference between $p_t(x)$ and $p_t(x_t)$ according to Eqn. (5)? Eqn. (5) depends on both $x_t$ and $x$.
   * $x_{0|t}^{(i)} \sim \mathcal{N}(x; x_t, \sigma_t^2 I)$ in Eqns. (7), (9), etc., but what does it mean by $x$? It'd be better understandable if it is written as $x_{0|t}^{(i)} \sim \mathcal{N}(x_{0|t}^{(i)}; x_t, \sigma_t^2 I)$ or simply $x_{0|t}^{(i)} \sim \mathcal{N}(x_t, \sigma_t^2 I)$.
   * Moreover, the notation $x_{s|t}^{(i)} \sim \mathcal{N}(x; x_s, (\sigma_t^2 - \sigma_s^2) I)$ in Eqns. (15), etc. is inconsistent unless $x_s$ is replaced by $x_t$. Is $x_s$ an errata?

   I believe that those issues arise as the authors always used $x_t$ instead of $x$. I understand that $x$ is a general variable but $x_t$ is a random variable generated by the SDE. E.g., Eqn. (5) does not need to depend on $x_t$ which the authors can replace to $x$ (if $x$ is not used in $\mathcal{N}(\cdot)$) and say that $p_t(x)$ is the (marginal?) distribution of $x_t$. Another issue with Eqn. (5) is that $\mathcal{N}(x; x_0, \sigma_t^2 I)$ becomes a Dirac delta when $t \to 0$ (as $\sigma_0 = 0$), but what is $\mathcal{N}(x_t; x_0, \sigma_t^2 I)$ when $t \to 0$? (both mean and variance converge to zero in this case).


* **Technical Points.**
   - Right above Eqn. (2), the authors said that the base distribution $p_1$ is known, but $p_t(x)$ (at $t = 1$) is a complex integration (5), e.g., not Gaussian. What does it mean by a *known* distribution $p_1$? In general, $p_0$ is not necessarily Gaussian, so that $p_t$ would be Gaussian but only in the limit $t \to \infty$, i.e., $p_1$ can be a complex distribution that is hard to be known.

   - $x_{s|t}^{(i)} \sim \mathcal{N}(x; x_s, (\sigma_t^2 - \sigma_s^2) I)$. Although the variance "$\sigma_t^2 - \sigma_s^2$" is smaller, the mean $x_s$ is still a random variable which indeed further increases the variance of $x_{s|t}^{(i)}$. Have the authors properly addressed it? If not, then the mathematical claim of BNEM would be just a result of not dealing with that missing variance, although the experiments show that BNEM would reduce the variance.

   - **Bi-level iterative scheme on pp. 4 and Fig. 1.** Although Akhound-Sadegh et al. (2024) and Midgley et al. (2023) used this technique, it is a potential issue that the replay buffer will have a different distribution if $\nabla E_\theta$ is not trained well/enough? How do you ensure that the distribution of the replay buffer $\mathcal{B}$ is the same as $\mu_\mathrm{target}$? (I haven’t followed those references though)…

   - **Approximations.** The authors have used equalities "$=$" instead of approximations "$\approx$". E.g. Eqns. (17) and (18) in Proposition 3.4 and the equality for the Bias[ . ] on pp. 6 of bootstrapping-once scenario are actually all approximations, according to their proof in the Appendix; the bound (11) in Proposition 3.1 comes with $\mathcal{O}(1/K)$ according to its proof. The same goes in the proofs (i.e., using $\approx$ instead of $=$; Eqns (28), (29), (42) need "$+ \mathcal{O}(1/K^2)$"). The authors need to carefully address this for rigorousness and further discuss those approximation preferably in the appropriate place of the main body.

   - How did the authors obtain the Taylor expansions (24) and (25)? Can you give some more details?

   - When Eq. (44) is transformed to Eq. (45), where is the variance $\sigma_t$ in Eq. (45)? In (41)--(49), is $x_t$, which is also an argument of $E_{\theta^*}(\cdot)$, already a random variable or just a general variable (like $x$)?

   - Why the bound in Proposition 3.1 is true with probability $1 - \delta$? I can't find any relevant proof in the Appendix.

   - When you apply techniques relevant to sub-Gaussians, can you give some more explanations in Appendix, with a proper references? E.g., right above Eq. (24), why $E_K$ is also sub-Gaussian? Also, for Eq. (26). Right above (40), "Therefore, $S_K$ is bounded by $M/c$, suggesting it is sub-Gaussian." --- how?

   - Right above Eq. (95), "When $K$ is large and $\sigma_s^2$ is small...": Is it $\sigma_t^2 - \sigma_s^2$? If not, how is $\sigma_s^2$ small when $s$ is close to 1?

   - How the approximation (95) is valid under the assumption, according to the definitions of $m_s$ and $E_K$?

# Other Comments
* When Propositions, Lemmas and Corollaries are referred, they are always named "Theorems". E.g., right below Proposition 3.1, it is said "Theorem 3.1 shows an error bound ...".
* The numbers of the Propositions and Corollaries and Equations in the Appendices are not matched with the original numbers. E.g., on page 15, Proposition 3.1 and Eq. (11) become Proposition B.1 and Eq. (19).
* In " ... the bias of BNEM (Eq. (20)) can be smaller than that of NEM (Eq. (18)) ... " on pp. 6, "Eq. (20)" seems an errata .
* I understood it, but it'd be good if the authors add more details about why the bound $\sigma_{t_{i+1}}^2 - \sigma_{t_i^2} \leq \beta/2$ is chosen on pp. 6, rather than $\sigma_{t_{i+1}}^2 - \sigma_{t_i^2} \leq \beta$.
* Can the authors explain why Proposition 3.1 is restricted to 1-dim. on pp. 5? What does it mean here by the dimension, and how is it different between 1-dim and n-dim?
* In the first line on pp. 5, an errata: vecinity $\to$ vicinity.
* Can we get any general insight from the results? The Boltzmann distribution (1) can be considered as just another expression of an arbitrary pdf since $\mathcal{E}$ is arbitrary. Or, is there any restriction on $\mathcal{E}$ for it to be an energy function?
* Right above Eq. (88), "$b_i$ are close to $0$ and concentrated at $m_b$. I believe it also means $m_b$ is close to $0$. If it does, please add one sentence for it.

---

> ### Author Response · Authors · 2025-12-30
> **Response to Reviewer Rcqo [1/2]**
>
> Dear Reviewer **Rcqo**,
>
> We thank you for their thoughtful and constructive feedback, as well as for your positive assessment of the contribution.
>
> We address all concerns in detail below and have revised the manuscript accordingly. All changes are marked in red in the updated version.
>
> > ## Confusing Notations
>
> We appreciate the reviewer pointing out the typos. In Eq(6), it is $\epsilon_t(x_t)$, not $\epsilon_t(x)$. And in Eq(15), it is an errata, where the mean of the Gaussian should be $x_t$ not $x_s$. We fixed these typos in the updated manuscript.
>
> In both Eq(5) and $N(x; x_t, \sigma_t^2I)$, the distributions converge to a delta function when $t\rightarrow 0$. The distinction is in the mean: the mean of Eq(5) is $x_0$ and the mean of the latter one is $x_t$.
>
> > ## Tractability of the base distribution
>
> While the target distribution $p_0$ is complex/intractable, the diffusion process ensures that $p_1$ becomes tractable.
>
>  - For the Variance Preserving (VP) case (used in [1]), parameters are chosen such that p1​ converges exactly to N(0,I).
>
>  - For the Variance Exploding (VE) case (which we utilize), we define $x_t​=x_0​+\sigma_t ​z$ where $z\sim N(0,I)$. As $\sigma_t$ becomes large, the convolution $p_1​=p_0​*N(0,\sigma_1^2​I)$ is dominated by the noise, meaning $p_1$​ converges approximately to $N(0,\sigma_1^2 ​I)$ regardless of the structure of p0​. This is a standard property in diffusion model literature (e.g., [1]), ensuring that we can sample from a "known" base distribution to initialize the reverse chain. We have clarified this approximation in the text.
>
> > ## On the variance
>
> There may be a slight misunderstanding regarding the variance we aim to reduce. While $x_t$​ is indeed a random variable (sampled from the SDE), it serves as the input location for the network. The "variance" we address in BNEM is the variance of the **regression target estimation** at that specific $x_t$.
>
>  - In standard NEM/iDEM, the target is a noisy estimate.
>
>  - In BNEM, we bootstrap to reduce the variance of this target given an $x_t$​. Therefore, while the sampling of $x_t$​ retains its inherent stochasticity, BNEM provides a cleaner learning signal for the network at those sampled locations, leading to better optimization.
>
> > ## On the bi-level iterative scheme
>
> This is an important point. However, a biased replay buffer does not necessarily lead to a biased energy/score model. The regression target (e.g., $\epsilon_t(x_t)$ in BNEM/NEM or $-\nabla \epsilon_t(x_t)$ in iDEM) is defined at location xt​. The replay buffer essentially determines the **weighting** of the loss function across the domain (i.e., where we minimize the error). As long as the replay buffer covers the support of the target modes (even with incorrect relative weights initially), the model will learn the correct local gradients/energies. As training progresses, the improved model effectively "corrects" the sampler, aligning the buffer distribution closer to the target.
>
> > ## Taylor expansion in Eq (24) and Eq (25)
>
> This result is given by standard logarithm of sub-Gaussian random variables. Let a sub-Gaussian random variable $X$ has mean $m$ and variance $v$, then $\log X$ has mean $log m - \frac{m^2}{2v}$ and variance $v/m^2$. These are standard results in statistics.
>
> > ## On Eq 44 -> Eq 45 and Eq (41)-(49)
>
> In equations (41)-(49), we are talking about the score at a specific location $x_t$, i.e. $x_t$ is a fixed variable instead of a random variable.
>
> For Eq (45), the author apologizes that there’s a typo: it should be $\epsilon_{0|t}^{(i)}\sim N(0, \sigma_t^2 I)$ but not $\epsilon_{0|t}^{(i)}\sim N(0, I)$
>
> > ## The bound given by proposition 3.1
>
> This result is derived using standard concentration inequalities (specifically Hoeffding’s or Chernoff bounds) followed by a triangle inequality application. We gave more details on the updated manuscript.
>
> > ## On the technical details of sub-gaussian random variables
>
> The relevant techniques applied are standard results in high-dimensional statistics (e.g., [2]).. The one above Eq (24) is the mean and variance of logarithm of sub-Gaussian variables. The one above Eq (26) is the concentration inequality of sub-Gaussian variables. The one above Eq (40) is that “if a random variable X is a summation of sub-Gaussian random variables and is bounded, then X is also sub-Gaussian”. The author doesn’t include their details as they are standard properties of sub-Gaussian variables. We will include more details if necessary.

---

> > ### Author Response · Authors · 2025-12-30
> > **Response to Reviewer Rcqo [2/2]**
> >
> > > ## Approximations
> >
> > We thank the reviewer for their close reading and for pointing out the need for greater mathematical rigor regarding the asymptotic terms. We agree that treating these relationships as strict equalities without quantifying the error was imprecise.
> > To address this, we have revised Proposition 3.1 and Proposition 3.3, along with their respective proofs, to explicitly include the asymptotic error terms (using Big-O notation).
> >
> > 1. In the Proof of Proposition 3.1 (Eq. 28 & 29): We have refined the derivation to include the O(1/K^2) residual that arises from the Taylor expansion of the bias. The revised steps now strictly quantify the approximation error:
> >
> > $$\|E_K(x_t, t)-\mathcal{E}_t(x_t)\| \leq \| E_K(x_t, t)-\mathbb{E}[E_K(x_t, t)]\| + \frac{v_{0t}(x_t)}{2m^2_{t}(x_t)K} + \mathcal{O}(1/K^2) \leq \sqrt{2\frac{v_{0t}(x_t)}{m^2_{t}(x_t)K}\log\frac{2}{\delta}} + \frac{v_{0t}(x_t)}{2m^2_{t}(x_t)K} + \mathcal{O}(1/K^2)$$
> >
> > It is worth noting that the bound is dominated by the $K^{-1/2}$ concentration term; the additional remainder introduced by the Taylor expansion of the bias is of strictly smaller order and therefore does not affect the resulting concentration rate.
> >
> > 2. In Proposition 3.3 (Eq. 42): We have revised the comparison between the variances of the score and energy estimators. Instead of a direct equality, we now present it as a ratio with an explicit error term, derived from the higher-order terms of the sub-Gaussian approximation:
> >
> > $$\frac{\text{Var}[S_K(x_t, t)]}{\text{Var}[E_K(x_t, t)]} = {4(1+\|\nabla\mathcal{E}_t(x_t)\|)^2}+\mathcal{O}\left(\frac{1}{K}\right)$$
> >
> > 3. Finally, regarding Proposition 3.4, we have opted to retain the current formulation without explicitly expanding the higher-order error terms. While we have rigorously quantified the $O(1/K)$ residuals in the foundational Propositions 3.1 and 3.3 as requested, propagating these terms into Proposition 3.4 results in an unwieldy expression that obscures the primary theoretical insight. Given that the underlying estimators are now rigorously bounded in the preceding propositions, we believe the current form of Proposition 3.4 correctly characterizes the **dominant behavior** of the system and suffices for the claims made in this section.
> >
> > > ## On typos above Eq (95)
> >
> > The author apologizes for the typos there. It is indeed $\sigma_t^2-\sigma_s^2$.
> >
> > > ## The validity of Eq (95)
> >
> > According to Eq (86), we have $\log m_s(x_s)\approx -\mathbb{E}[E_K(x_s, s)]+ O(1/K)$. The error is bounded linearly by $K$, justifying the approximation.
> >
> > > ## On the numbers and names of propositions and corollaries
> >
> > We apologize for the rendering issues. We have standardized all references to Propositions, Lemmas, and Corollaries, and corrected the numbering mismatch between the main text and Appendix.
> >
> > > ## The chosen bounds for variance control in BNEM
> >
> > $\sigma_{i+1}^2-\sigma_i^2\leq \beta/2$ is chosen because, in practice, we sample $t\in[t_i, t_{i+1}]$ and $s\in[t_{i-1}, t_i]$, which means the maximum variance difference is $\sigma_{i+1}^2-\sigma_{i-1}^2$. Therefore, choosing $\sigma_{i+1}^2-\sigma_i^2\leq \beta/2$ ensures $\sigma_{i+1}^2-\sigma_{i-1}^2 \leq \beta$
> >
> > > ## On the dimensionality of proposition 3.3
> >
> > The restriction allows for a simplified analysis because the energy is a scalar, whereas the score (gradient) is a vector in high-dimensional spaces. In 1D, the relationship between the energy derivative and the score is diwrect, simplifying the bounds. The intuition extends to higher dimensions, but the scalar bounds are tighter and easier to prove.
> >
> > > ## General insight
> >
> > We thank the reviewer for this insightful comment. You are correct that the Boltzmann distribution (Eq. 1) is a general representation that can describe any probability density. However, while we do not restrict the functional form of $\mathcal{E}(x)$ in practice, our theoretical analysis (Propositions 3.1–3.4) imposes a regularity condition: we assume that the random variable $\exp(−\mathcal{E}(x))$ is sub-Gaussian with respect to the sampling distribution. This restriction is crucial for deriving our non-asymptotic bounds, as it ensures that the Monte Carlo estimators do not suffer from heavy-tailed variance
> >
> > ---
> >
> > ## End of Response
> >
> > We thank the reviewer again for their constructive comments, which have explicitly improved the quality of presentation, mathematical rigor, and clarity of our theoretical analysis. We have updated the manuscript to reflect these corrections and hope that our revisions and explanations satisfactorily address the concerns raised.
> >
> > ---
> >
> > [1] Song, Yang, et al. "Score-Based Generative Modeling through Stochastic Differential Equations." International Conference on Learning Representations, 2021
> >
> > [2] Wainwright, Martin J. High-Dimensional Statistics: A Non-Asymptotic Viewpoint. Cambridge University Press, 2019.

---

> > ### Comment · Reviewer_Rcqo · 2026-02-10
> > **Reply to Response to Reviewer Rcqo [1/2]**
> >
> > I apologise the delay due to the late response. Here are my concerns after re-reviewing the manuscript with your response and revision. I’ll respond your answer without further delay this time.
> >
> > # Confusing Notations
> > Still, $\mathcal{N}(x_t; x_0, \sigma_t^2)$ in the integral of Eq. (5) does not converge to the Dirac delta as $t \to 0$. Where does $(x_t - x_0)^2 / \sigma_t^2$ converge when $x_t \to x_0$ and $\sigma_t \to 0$? The problem is that the authors abused the notation $x_t$, which has a specific meaning (i.e., the state at time $t$ generated by an SDE), rather than using general variable $x$. E.g., if I were an author, I'd write Eq. (5) as
> >
> > $$
> >   p_t(x) \propto \int \exp (-\mathcal{E}(x_0)) \mathcal{N}(x ; x_0, \sigma_t^2 I) d x_0
> >   = \mathbb{E}_{z \sim \mathcal{N}( z ; x, \sigma_t^2 I)}[\exp (-\mathcal{E}(z))]
> > $$
> >
> > (or $q_t(x | x_0)$ can be used here and there to simplify the notation). The definition of $p_t$ is then general (previously it was valid only at the trajectories $x_t$) and does not cause the problem mentioned above when $t \to 0$. The issues due to the abuse of $x_t$ actually exist throughout the paper, which confuses the reader. The authors also abused the notation $x$ when defining the distribution, e.g., $x_{0|t}^{(i)} \sim \mathcal{N}( x ; x_t, \sigma_t^2 I)$, which doubles the confusion. $x_{0|t}^{(i)} \sim \mathcal{N}( x_{0|t}^{(i)} ; x_t, \sigma_t^2 I)$ is correct and if it's too complex, it can be written simply as $x_{0|t}^{(i)} \sim q_t( x_{0|t}^{(i)} | x_t)$ or the standard notation $x_{0|t}^{(i)} \sim \mathcal{N}( x_t, \sigma_t^2 I)$ interchangeably, e.g., as Song et al. (2021) Denoising Diffusion Implicit Models. Why not using the simpler notation $q_t( x | x_0)$ that has already been defined? Also note that it is defined also with abuse of $x_t$ in the manuscript.
> >
> > # Tractability of the base distribution
> >
> > I disagree with it. (1) Neither VP nor VE converge exactly to a Gaussian distribution. The state variance of VP in [1] has a closed form expression, but the distribution is not proven Gaussian (only the distribution $p_t( x_t | x_0)$ conditional on the initial state is Gaussian) --- see Appendix B in [1]; for the VE, $p_1$ is Gaussian only approximately, as the authors mentioned in the reply. So, for any case, $p_1$ is the known base distribution but "only approximately".
> >
> > # On Eq 44 -> Eq 45 and Eq (41)-(49)
> >
> > "In equations (41)-(49), ... $x_t$ is a fixed variable instead of a random variable.": Then, it'd be better to use a general variable notation $x$ rather than $x_t$ as discussed above.
> >
> > # Taylor expansion in Eq (24) and Eq (25) & Sub-Gaussians...
> >
> > I appreciate the explanation, but I believe that it is beneficial to write those detailed logical explanation in the Appendix. Sub-gaussian techniques would be standard in stats fields, but not every research in ML are from stats background; I still do not understand how the first order Taylor expansion (24) and (25) are obtained.
> >
> > # On the bi-level iterative scheme
> > I cannot agree with the claim. In my understanding (like in DQN in reinforcement learning), replay buffer stores the data, $x_0$ in this case, and the algorithm samples uniformly among them, not necessarily weights of the loss function. I.e., replay buffer $\mathcal{B}$ itself defines the distribution of $x_0$ as indicated in Algorithm 1 (line 4) and Algorithm 2 (line 2). However, it is updated in the outer loop by Algorithm 1 (line 2) for the new $\theta$. That is, $\mathcal{B}$ indeed affects the energy function parameter $\theta$, and vice versa.
> >
> > "As training progresses, the improved model effectively "corrects" the sampler, aligning the buffer distribution closer to the target." -- is this a proven result or validated only experimentally? Including this, I'm happy to see any clarification regarding the replay buffer $\mathcal{B}$ and the argument $\mathrm{dist}(\mathcal{B}) \approx \mu_\mathrm{target}$.
> >
> > # The bound given by proposition 3.1
> > I cannot find the updated content for this. Can you refer to the spot where Hoeffding’s or Chernoff bound is mentioned, followed by a triangle inequality?

---

> > > ### Author Response · Authors · 2026-02-14
> > > **Response to Reviewer Rcqo**
> > >
> > > Dear reviewer,
> > >
> > > Thanks for the comments, we would like to answer your questions as follows:
> > >
> > > ## Regarding proposition 3.1
> > >
> > > 1. Firstly for the Taylor expansion used for estimating the mean and variance of Ek, as eq 24 and 25 in the revised manuscript, they are following the standard result in statistics, where the mean and variance of "logarithm of a sub-gaussian variable" can be quantified by the mean and variance of the sub-gaussian variable itself. We add few lines to explain the derivation and include the reference for this technique.
> > >
> > > 2. Secondly, on the inequalities. We include why the first inequality exists, by using the definition of sub-Gaussianess.
> > >
> > > 3. We agree that the results are approximately true, we claimed them in the revised manuscript.
> > >
> > >
> > > ## Notational issue
> > >
> > > We changed the notation accordingly, see eq 7, 9, 15 etc.
> > >
> > > ## Validation of Eq 95 (now is 97 in the revised manuscript)
> > >
> > > We insist that it is valid. From Corollary 1, we know that $E_K(x,s)$ has small bias and variance (scaling with $1/K$). Let $E_K(x, s)=E_s(x)+\delta$, where $\delta$ represents the estimation error. Since $\delta$ is small with high probability (due to the sub-Gaussian assumption):
> > >
> > > $$\exp(-E_K(x, s))=\exp(-E_s(x))\exp(-\delta)\approx m_s(x)(1-\delta)$$
> > >
> > > When we sum over K samples, these small errors are averaged, resulting in eq 98.
> > >
> > > ## On the bi-level iterative scheme
> > >
> > > We agree that the reply buffer affects the optimization process. However, it doesn't affect the optimality, since the optimality doesn't depend on the distribution defined by the replay buffer. In fact, the pointwise regression target is, ideally, the exact noised energy/score, while the replay buffer tells where should we evaluate the loss. Intuitively speaking, the iterative training is fulfilling the whole landscape, which contains the support of the target distribution, and therefore could iteratively improve the sampler quality. But at least, in this paper, the effectiveness of the replay buffer is shown in practice, not theoretically.
> > >
> > > We thank the reviewer again for their constructive comments, which have explicitly improved the quality of presentation, mathematical rigor, and clarity of our theoretical analysis. We have updated the manuscript to reflect these corrections and hope that our revisions and explanations satisfactorily address the concerns raised.

---

> ### Comment · Reviewer_Rcqo · 2026-02-10
> **Reply to Response to Reviewer Rcqo [2/2]**
>
> I apologise the delay due to the late response. Here are my concerns after re-reviewing the manuscript with your response and revision. I’ll respond your answer without further delay this time.
>
> # Approximations
>
> 1. It'd be great if you include the following Taylor series expansion to the proofs:
>
> $|E_K(x_t, t)-\mathcal{E}_t (x_t)|$
>
> $\displaystyle \leq | E_K(x_t, t)-\mathbb{E}[E_K(x_t, t)]| + \frac{v_{0t}(x_t)}{2m^2_{t}(x_t)K} + \mathcal{O}(1/K^2)
> \leq \sqrt{2\frac{v_{0t}(x_t)}{m^2_{t}(x_t)K}\log\frac{2}{\delta}} + \frac{v_{0t}(x_t)}{2m^2_{t}(x_t)K} + \mathcal{O}(1/K^2)
> $
>
> But, can you add/explain some more details about this Taylor series expansion (e.g. more steps of inequalities)?
>
> 2. In Eq. (42), the big O term $\mathcal{O}(1/K^2)$ is in both the numerator and the denominator. In this case, can we really claim that the var-ratio has $\mathcal{O}(1/K)$ term? It seems that the equality should be true *only approximately* without Big O, or the derivation needs to be revised.
>
> 3. The equations in pp. 6 including those of Proposition 3.4 must be true *only approximately*. First, quantifying the big O terms in Eq. (42) of Proposition 3.3 has an issue as mentioned above. Second, even if the residual big O terms are rigorously quantified, the math should be precise. The Bias and Var can be characterised by those approximations (why not?); it is possible to use $\approx$ for the approximate equalities and justify, discuss and explain it why in the following paragraphs (or appendices if the space matters) in relation to big O terms.
>
> # Validity of Eq. (95)
>
> Eq. (85) -- $\mathrm{log}$ $m_s(x_s) = - \mathbb{E} [E_K(x_s, s)] + O(1/K)$ -- does not justify Eq. (95). It should be true without $\mathrm{log}$ that $m_s(x_s) = - \mathbb{E} [E_K(x_s, s)] + O(1/K)$, not (85). Also note that the log function is extremely sensitive near zero.
>
> # The chosen bounds for variance control in BNEM
>
> What I meant is to just give a little hint, e.g., by writing "such that $\sigma_{t_{i+1}}^2 - \sigma_{t_i}^2 \leq \beta/2$ *(which ensures $\sigma_{t}^2 - \sigma_{s}^2 \leq \beta$ for any $t, s \in [t_i, t_{i+1}]$)*".

---

> ### Author Response · Authors · 2026-02-14
> **Response to Reviewer Rcqo**
>
> ## O(1/K^2) terms in both numerator and denominator in eq (42) (now eq 45 in the revised manuscript)
>
> We apologise that we mistakenly commented the explanation for this step. We now added back this explaination again:
> "
> Step (i) follows from the asymptotic expansion of the ratio. By factoring out the leading order term $1/K$ from both the numerator and denominator, the quotient converges to the ratio of the coefficients with a residual error of order $\mathcal{O}(1/K)$.
> "

---

### Decision · Action_Editor_s4bV · 2026-03-04

**Recommendation:** Accept with minor revision

**Audience:**

Yes

**Audience Explanation:**

Learning an IID sampler from energy function is a long-standing challenge that is the fundamental task underlying various problems including molecular science, Bayesian inference, and solving inverse problems. The paper pushed forward the potential of using diffusion models for this purpose. Although the technical route is not ground-breaking given the work by Akhound-Sadegh et al. (2024), the authors answered a relevant question about the advantage of using the conservative form of score to learn from the given energy function, especially in addressing the challenge that the variance of the estimation of the learning target increases with the diffusion time step. It is a pity that the paper does not seem to have extended the scale of applicability of learning a diffusion sampler for Boltzmann sampling, though. Nevertheless, it still contributed relevant and valuable findings and inspirations in this field.

**Claims And Evidence:**

Yes

**Claims Explanation:**

The paper proposes an approach for IID sampling for the Boltzmann distribution by learning a diffusion model from the given energy function (Hamiltonian) that defines the Boltzmann distribution. The key novelty/claim is that the authors pointed out that learning the energy/conservative form of the score function at each diffusion time step, could benefit from a better sample efficiency / lower variance of the Monte Carlo estimation of the learning objective than directly learning the vector-field form of the score, and it also enables bootstrapping from the learned energy model at a smaller time step / noise scale for learning the energy function at a larger scale. These claims are clearly and formally described by statistical analysis and experimental demonstrations.

All reviewers acknowledged the relevance, novelty, and a clear presentation. Reviewer Rcqo mentioned a few concerns regarding technical validity after thorough reads over the whole paper. I would perceive these points as minor, as the proposed method learns intermediate-time-step energy functions from the given terminal energy function and the reference data distribution does not alter optimality theoretically. Nevertheless, the authors could make this point clearer and more explicit in the paper, and mention its practical impact as there would still be an exploration problem in practice, which has long obscured learning from energy over learning from data directly.